# The *Candida albicans* transcription factor Cas5 couples stress responses, drug resistance and cell cycle regulation

Jinglin L. Xie [1], Longguang Qin[2], Zhengqiang Miao[2], Ben T. Grys [1,3], Jacinto De La Cruz Diaz[4], Kenneth Ting[1], Jonathan R. Krieger[5], Jiefei Tong[6], Kaeling Tan[2], Michelle D. Leach[1,7], Troy Ketela[1], Michael F. Moran[1,5,6], Damian J. Krysan[4,8], Charles Boone[1,3], Brenda J. Andrews[1,3], Anna Selmecki[9], Koon Ho Wong[2], Nicole Robbins[1] & Leah E. Cowen[1]

The capacity to coordinate environmental sensing with initiation of cellular responses underpins microbial survival and is crucial for virulence and stress responses in microbial pathogens. Here we define circuitry that enables the fungal pathogen *Candida albicans* to couple cell cycle dynamics with responses to cell wall stress induced by echinocandins, a front-line class of antifungal drugs. We discover that the *C. albicans* transcription factor Cas5 is crucial for proper cell cycle dynamics and responses to echinocandins, which inhibit β-1,3-glucan synthesis. Cas5 has distinct transcriptional targets under basal and stress conditions, is activated by the phosphatase Glc7, and can regulate the expression of target genes in concert with the transcriptional regulators Swi4 and Swi6. Thus, we illuminate a mechanism of transcriptional control that couples cell wall integrity with cell cycle regulation, and uncover circuitry governing antifungal drug resistance.

[1] Department of Molecular Genetics, University of Toronto, Toronto, ON, Canada M5G 1M1. [2] Faculty of Health Sciences, University of Macau, Macau SAR 999078, China. [3] Donnelly Centre for Cellular and Biomolecular Research, University of Toronto, Toronto, ON, Canada M5S 3E1. [4] Department of Microbiology and Immunology, University of Rochester, Rochester, NY 14642, USA. [5] The Hospital for Sick Children, SPARC Biocentre, Toronto, ON, Canada M5G 0A4. [6] The Hospital for Sick Children, Program in Cell Biology, Peter Gilgan Centre for Research and Learning, Toronto, ON, Canada M5G 0A4. [7] Aberdeen Fungal Group, School of Medical Sciences, Institute of Medical Sciences, University of Aberdeen, Foresterhill, Aberdeen AB252ZD, UK. [8] Department of Pediatrics and Microbiology/Immunology, University of Rochester, Rochester, NY 14642, USA. [9] Department of Medical Microbiology and Immunology, Creighton University School of Medicine, Omaha, NE 68178, USA. Correspondence and requests for materials should be addressed to L.E.C. (email: leah.cowen@utoronto.ca)

The fungal kingdom encompasses diverse species, including a minority that have a devastating impact on human health. One of the most pervasive fungal pathogens of humans is *Candida albicans*[1, 2], which is a commensal on the skin and mucosal surfaces of up to 60% of healthy individuals[3]. As an opportunist, *C. albicans* can exploit a decline in host immunity or an imbalance in the host microbiome, leading to diverse pathologies such as oral thrush, vaginal candidiasis, or life-threatening bloodstream infections with mortality rates of ~40%[4, 5]. *C. albicans* thrives as a human pathogen in part due to its ability to evade host immunity by switching between yeast and filamentous morphologies, as well as due to its capacity to withstand the hostile host environment by activating robust stress responses[6]. The emerging paradigm is that *C. albicans* stress response pathways are not only critical for adaptation to host conditions, but they also enable fungal virulence and drug resistance[7–11].

The emergence of resistance to the limited arsenal of antifungal drugs impedes the effective treatment of systemic infections[12–14]. A poignant example is the evolution of resistance to the only new class of antifungal to be approved in decades, the echinocandins[15, 16]. Echinocandins block β-1,3-glucan biosynthesis in the fungal cell wall via inhibition of the glucan synthase Fks1, thereby compromising cell wall integrity. The most common mechanism of echinocandin resistance involves mutations in the drug target *FKS1*[17, 18]; however, resistance phenotypes are also modulated by cellular stress responses[8, 9, 11]. Targeting core hubs in cellular circuitry that control responses to stress, such as the molecular chaperone Hsp90 or protein phosphatase calcineurin, has emerged as a powerful strategy to enhance antifungal activity against diverse fungal pathogens[8, 16]. A challenge in targeting Hsp90 or calcineurin in antifungal drug development is that they are conserved and important regulators in the human host.

An expanded repertoire of potential targets is provided by effectors downstream of these cellular regulators. *C. albicans* mobilizes diverse stress response programs through the action of transcription factors. For example, in response to cell membrane and cell wall stress, the transcription factor Crz1 is activated by calcineurin, leading to the induction of calcineurin-dependent genes[19, 20]. Another example from the model yeast *Saccharomyces cerevisiae* is the cell wall stress-dependent activation of the transcription factor Rlm1 by the MAP kinase Mpk1[21]. Although Rlm1 is the main transcriptional regulator of cell wall stress responses in *S. cerevisiae*, its function is not conserved in *C. albicans*[22]. In *C. albicans*, the zinc finger transcription factor, Cas5, serves as a key transcriptional regulator of responses to cell wall stress[22]. Cas5 lacks an ortholog in *S. cerevisiae* and most other eukaryotes, and the mechanism by which it is regulated remains enigmatic.

Activation of stress responses can induce diverse physiological changes, including modulation of cell cycle progression and remodeling of cell wall architecture[23–27]. The most well characterized stress response pathway involved in cell cycle regulation is controlled by the MAP kinase Hog1[26]. In response to osmotic stress, Hog1 mediates a transient cell cycle arrest to enable cellular adaptation[26]. Multiple stress response pathways coordinate cell wall remodeling in response to environmental perturbations, including heat shock[27], osmotic stress[28], and cell wall stress[29]. However, little is known about whether cell cycle progression and cell wall remodeling are coordinated in response to stress in *C. albicans*, and whether there is a central signaling pathway that integrates these fundamental aspects of cellular biology.

Here, we harness genome-wide approaches to elucidate the mechanism by which Cas5 orchestrates transcriptional changes in response to cell wall perturbation. We identify an unexpected role for Cas5 in coupling cell cycle dynamics to cell wall stress responses, and determine that Cas5 regulates distinct transcriptional programs under basal and stress conditions. We discover that Cas5 is activated by the type I protein phosphatase Glc7 in response to cell wall stress, and it regulates cell wall homeostasis in part through its interaction with Swi4-Swi6 cell cycle box-binding factor (SBF) complex members, Swi4 and Swi6. Finally, we describe a role for Cas5 in orchestrating nuclear division. Our work provides insight into cellular reprogramming in response to cell wall stress, and establishes regulatory circuitry that couples stress response, cell cycle regulation, and drug resistance.

## Results

**Cas5 is a biologically responsive transcriptional regulator.** Cas5 was originally identified in a genetic screen for transcription factor mutants that are hypersensitive to the echinocandin caspofungin, and microarray analysis implicated Cas5 in the regulation of genes important for cell wall integrity in response to cell wall stress[22]. However, the role of Cas5 under basal conditions remained elusive. To further understand Cas5 function under basal conditions, we sought to explore the impact of Cas5 on regulation of global transcriptional profiles in rich medium. To do so, we used chromatin immunoprecipitation (ChIP) of RNA polymerase II (PolII) coupled to sequencing (ChIP-Seq) with a wild-type strain and *cas5Δ/cas5Δ* mutant. RNA PolII occupancy has been found to serve as a more precise measure of transcriptional regulation than steady state transcript levels, given that transcript levels depend on RNA turnover in addition to transcription factor activity and RNA PolII-mediated transcription[30]. As RNA PolII recruitment corresponds to transcriptional activity, an increase in RNA PolII occupancy indicates upregulation of gene expression, whereas a decrease in RNA PolII occupancy indicates downregulation[30, 31]. Further, strong correlation in RNA PolII binding between our biological replicates supported the reproducibility of this method (Supplementary Fig. 1). We identified 329 genes that had increased RNA PolII occupancy, and 275 genes that had reduced PolII occupancy in the *cas5Δ/cas5Δ* mutant as compared with the wild-type strain under standard growth conditions (Supplementary Data 1). To identify the physiological roles of Cas5, we subjected these gene sets to pathway analysis using Gene Ontology (GO). The sets that had reduced RNA PolII occupancy upon deletion of *CAS5* were significantly enriched in genes with functions in diverse processes, including metabolic processes and interaction with host (Fig. 1a and Supplementary Data 1). In contrast, the gene set that had increased RNA PolII occupancy in a *cas5Δ/cas5Δ* mutant was enriched in genes with functions associated with rRNA processing, respiration, and amino acid biogenesis (Fig. 1a and Supplementary Data 1). We also employed Kyoto Encyclopedia of Genes and Genomes (KEGG) analysis as a secondary approach to identify genes and pathways affected by deletion of *CAS5*. Consistent with our GO term analysis, many genes with RNA PolII-binding profiles that were affected by Cas5 were associated with ribosome biogenesis and metabolic pathways (Fig. 1b and Supplementary Data 1). KEGG analysis also identified a signature of differentially bound genes with functions in meiosis, cell cycle, and DNA replication (Fig. 1c and Supplementary Data 1). Thus, Cas5 governs diverse biological responses under basal conditions.

Next, we explored the impact of Cas5 on global responses to cell wall stress. We mapped genome-wide RNA PolII occupancy during caspofungin treatment in wild-type and *cas5Δ/cas5Δ* strains. In a wild-type strain, 546 genes had distinct RNA PolII-binding profiles in response to caspofungin treatment, with increased binding for 294 genes and reduced binding for 252 genes (Supplementary Data 2). As expected, many genes with

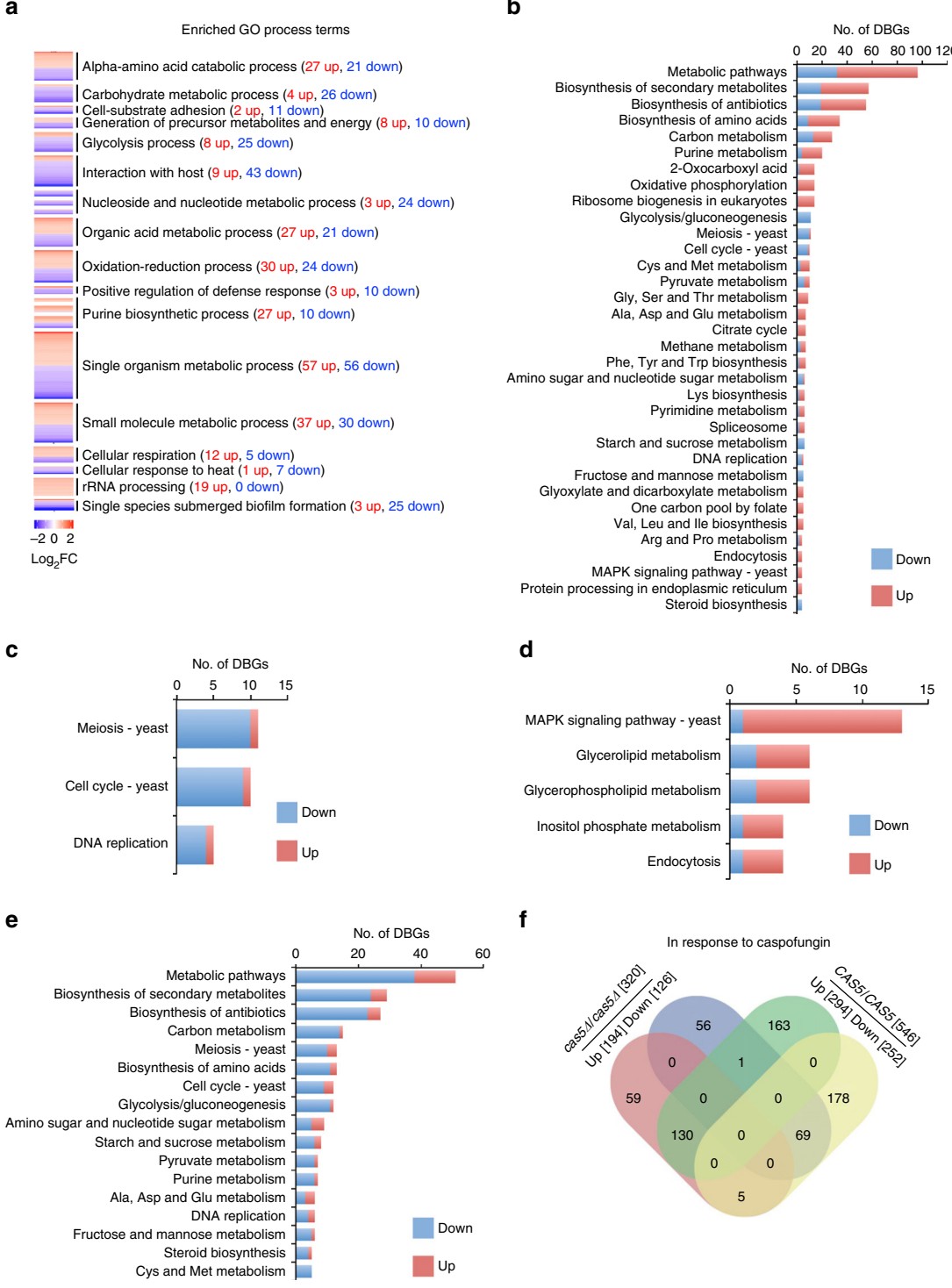

**Fig. 1** Cas5 has a profound impact on RNA PolII binding at genes implicated in cell wall and cell cycle processes under both basal and cell wall stress conditions. **a** Heat maps showing the genes with increased (*red*) and decreased (*blue*) RNA PolII binding in a *cas5Δ/cas5Δ* mutant relative to wildtype. Enriched GO processes are indicated, and were clustered using the DAVID Gene Functional Classification Tool. **b** Bar chart showing the number of genes differentially bound by PolII (*DBGs* differentially bound genes), with increased binding in *red* and decreased binding in *blue*, in a *cas5Δ/cas5Δ* mutant relative to wildtype based on their assigned KEGG pathways. KEGG pathways with four or more genes assigned are shown. **c** Bar chart showing the number of genes differentially bound by PolII, with increased binding in *red* and decreased binding in *blue*, in a *cas5Δ/cas5Δ* mutant relative to wildtype belonging to select KEGG pathways involved in cell cycle and related processes. **d**, **e** Bar charts showing the number of differentially bound genes involved in the indicated physiological pathways upon caspofungin treatment, with increased binding in *red* and decreased binding in *blue*, in a wild-type strain in response to caspofungin. KEGG pathways were grouped according to the ratio of genes with increased PolII binding to genes with decreased PolII binding in response to caspofungin, with **d** enriched for genes with increased PolII binding and **e** enriched for genes with decreased PolII binding. KEGG pathways with five or more genes assigned are shown. **f** Venn diagram depicting number of genes differentially bound by RNA PolII in response to caspofungin in the wild-type reference (increased, *green*; decreased, *yellow*) and *cas5Δ/cas5Δ* mutant (increased, *red*; decreased, *blue*) strains. See Supplementary Data 1–4 for full data sets

increased RNA PolII binding upon caspofungin exposure were important for tolerating cell wall stress (Fig. 1d and Supplementary Data 2), whereas genes with reduced RNA PolII binding were involved in diverse biological processes, including cell cycle progression (Fig. 1e and Supplementary Data 2). Strikingly, comparison of genes that were differentially bound in response to caspofungin between the wild-type strain and the *cas5Δ/cas5Δ* mutant, revealed that >60% of caspofungin-responsive genes were dependent on Cas5. Specifically, 163 of the 294 genes with increased PolII occupancy in response to caspofungin exposure and 178 of the 252 genes with reduced occupancy were dependent on Cas5 (Fig. 1f and Supplementary Data 2–4). These findings

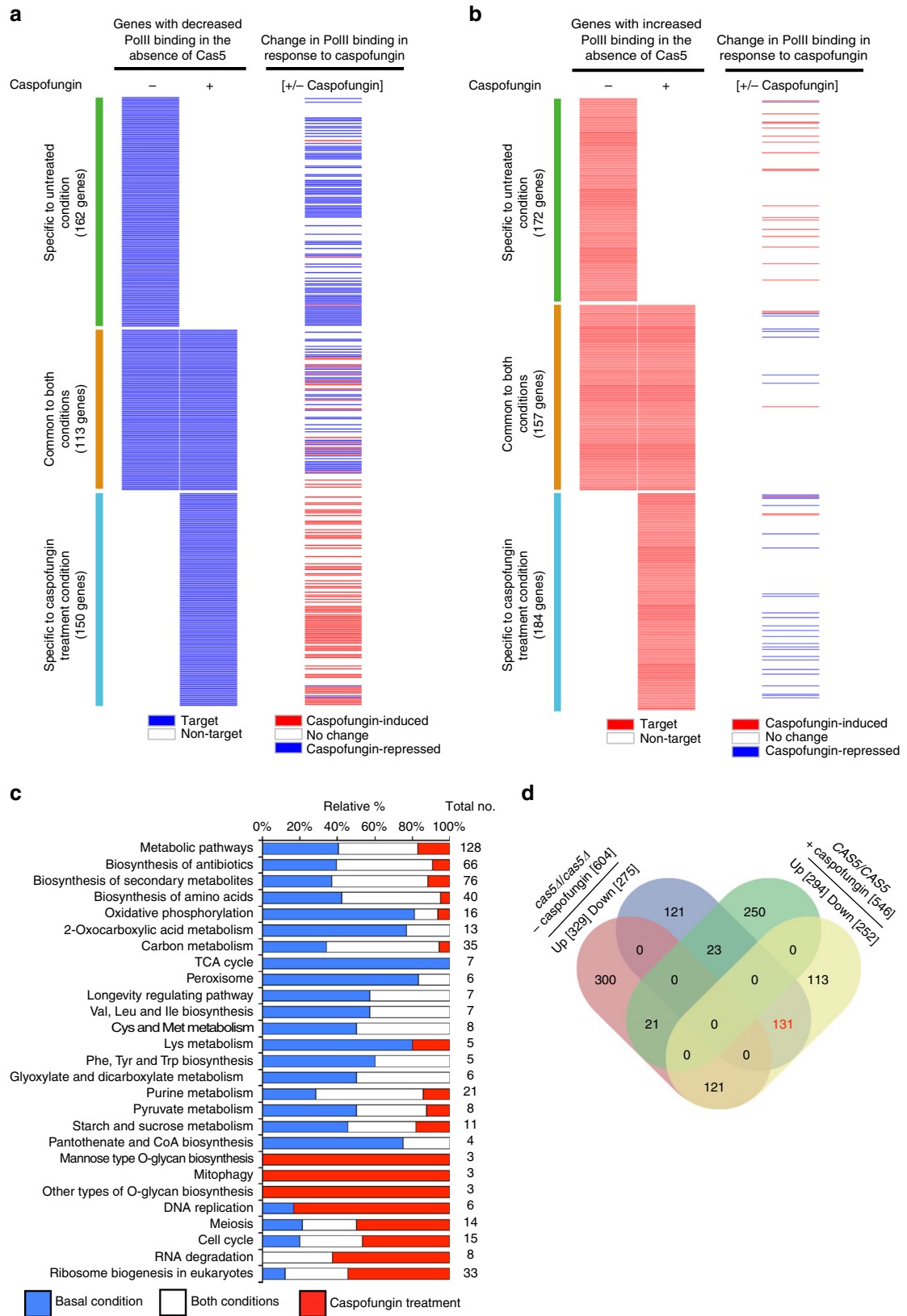

suggest that Cas5 has a profound impact on global transcriptional responses to cell wall stress.

Finally, we focused on those genes with Cas5-dependent differences in RNA PolII binding under basal and cell wall stress conditions. Strikingly, only 28% of genes with Cas5-dependent differences in RNA PolII binding were common to both untreated and caspofungin treatment conditions (Fig. 2a and Supplementary Data 1, 3 and 4). The Cas5-dependent genes specific to each condition had different physiological functions (Fig. 2c and Supplementary Data 4). Our analysis revealed a major overlap of genes that had Cas5-dependent increased RNA PolII occupancy under basal conditions with those that had reduced RNA PolII occupancy in a wild-type strain in response to caspofungin treatment (Fig. 2d and Supplementary Data 5), suggesting that caspofungin impedes Cas5-mediated expression of its basal-specific targets. Collectively, we identified hundreds of new Cas5-dependent transcriptional events under basal and stress conditions, implicating Cas5 in governing distinct transcriptional programs in response to different environmental conditions.

**Cas5 is a transcriptional regulator of cell wall homeostasis.** Given that Cas5 is known to regulate cell wall integrity[22], we sought to further evaluate the impact of Cas5 on cell wall homeostasis. We observed that Cas5 has a profound impact on RNA PolII binding to genes with cell wall functions; GO terms from process ontology revealed that 23 of the 163 genes with Cas5-dependent increases in RNA PolII binding in response to caspofungin encode proteins involved in cell wall organization and biogenesis (P-value $7.45e^{-06}$, Bonferroni Correction) (Supplementary Data 3). Among these genes were WSC1 and WSC2, which encode transmembrane sensors that respond to cell wall perturbations, yet have not been previously linked to Cas5. We confirmed the induction of these transcripts in response to caspofungin was dependent on Cas5 by quantitative RT-PCR (qRT-PCR) (Fig. 3a). We next evaluated the impact of Cas5 on tolerating cell wall stress in a wild-type strain and an echinocandin-resistant mutant. We measured susceptibility of a cas5Δ/cas5Δ mutant to two cell wall perturbing agents, caspofungin and calcofluor white. The cas5Δ/cas5Δ mutant was hypersensitive to both cell wall stressors, and the hypersensitivity phenotype was complemented by re-introduction of a wild-type allele (Fig. 3b). Given that the most common mechanism of resistance to echinocandins is mutations in the drug target gene, FKS1, we tested whether Cas5 also enables Fks-dependent caspofungin resistance. Homozygous deletion of CAS5 reduced resistance of a strain carrying the $Fks1^{F641S}$ substitution, suggesting Cas5 is a key regulator of target-mediated echinocandin resistance (Fig. 3b). Thus, Cas5 regulates cell wall stability during periods of cell wall stress, with a profound impact on drug resistance.

Next, we assessed if Cas5 regulates the expression of genes important for cell wall homeostasis in the absence of cell wall stress. We found that 110 out of the 275 genes that have reduced RNA PolII binding in the cas5Δ/cas5Δ mutant under basal conditions were associated with cell membrane, cell wall, and cell periphery GO components (Supplementary Data 1). Thus, we predicted Cas5 enables cell wall stability in the absence of cell wall stress, such that deletion of CAS5 would induce cell wall stress under basal conditions. To determine if a cas5Δ/cas5Δ mutant is intrinsically defective in cell wall architecture, we monitored phosphorylation of the terminal cell wall integrity MAP kinase, Mkc1, under basal conditions[32]. Compared with a wild-type control, the cas5Δ/cas5Δ mutant had elevated levels of Mkc1 phosphorylation under basal conditions (Fig. 3c and Supplementary Fig. 2a). Moreover, expression of RLM1, a transcription factor downstream of Mkc1, was upregulated in the cas5Δ/cas5Δ mutant under basal conditions (Fig. 3d), suggesting strains lacking Cas5 experience intrinsic cell wall stress. Our results indicate Cas5 maintains cell wall homeostasis under basal conditions and during cell wall stress.

**Cas5 is a transcriptional regulator of cell cycle dynamics.** We observed an intriguing role of Cas5 on governing RNA PolII binding at genes important for cell cycle, meiosis, and DNA replication (Figs. 1c and 2c), suggesting Cas5 regulates cell cycle dynamics under basal and cell wall stress conditions. If this were the case, the expression pattern of cell cycle regulated genes would be perturbed in the absence of Cas5. Using the Candida cell cycle database[33], we determined that 56 out of the 604 genes that had Cas5-dependent RNA PolII binding under basal conditions, were genes whose expression peaked during the cell cycle (Supplementary Data 6). Of these, over 75% (43 out of 56) had reduced RNA PolII binding in the cas5Δ/cas5Δ mutant as compared with the wild-type strain (Fig. 4a and Supplementary Data 6). Next, we analyzed the class of genes involved in cell cycle that had reduced RNA PolII binding upon caspofungin treatment (Fig. 1e). Among these genes were all six factors of the mini-chromosome maintenance (MCM) complex (Fig. 4b), which has important roles in the formation of the pre-replicative complex (pre-RC) prior to DNA replication[34]. Reduced RNA PolII binding at genes encoding MCM complex components was dependent on Cas5 (Fig. 4b). As expected, qRT-PCR confirmed that reduced expression of representative genes, MCM2 and MCM3, was also dependent on Cas5 (Fig. 4c). Finally, we classified genes that were differentially bound by RNA PolII in response to caspofungin based on their peak expression throughout the cell cycle, once again using the Candida cell cycle database[33]. In the wild-type strain, genes with increased RNA PolII binding in response to caspofungin were predominantly G1-specific, and those with reduced RNA PolII binding were predominantly M phase-specific

**Fig. 2** Cas5 regulatory network is rewired in response to caspofungin treatment. **a**, **b** Heat maps comparing Cas5-dependent genes with caspofungin-regulated genes. **a** The left heat map shows the overlap of genes with Cas5-dependent increased RNA PolII binding with (+) and without (−) caspofungin treatment. Cas5-dependent genes are indicated by a *solid blue line*. The *right* heat map indicates those genes with increased binding (*red*) and decreased binding (*blue*) in a wild-type strain upon caspofungin treatment. **b** The *left* heat map shows the overlap of genes with Cas5-dependent reduced RNA PolII binding with (+) and without (−) caspofungin treatment. Cas5-dependent genes are indicated by a *solid red line*. The *right* heat map indicates those genes with increased (*red*) or decreased (*blue*) RNA PolII binding in a wild-type strain upon caspofungin treatment. **c** Bar graph showing the relative percentage of Cas5-dependent genes specific to basal conditions (*blue bars*), specific to caspofungin treatment (*red bars*), or common to both conditions (*white bars*) in the indicated KEGG pathways. The total number of genes belonging to each KEGG pathway are indicated. Only selected KEGG pathways are shown based on number of genes enriched for PolII binding either under basal or caspofungin conditions. See Supplementary Data 4 for complete data set. **d** Venn diagram depicting number of genes differentially bound in a cas5Δ/cas5Δ mutant relative to wildtype under basal conditions (increased, *red*; decreased, *blue*) and genes responding to caspofungin treatment in a wild-type strain (increased, *green*; decreased, *yellow*). The 131 genes highlighted in *red* text represent the substantial fraction of the genes with Cas5-dependent increase in RNA PolII binding under basal conditions that have reduced PolII binding in response to caspofungin

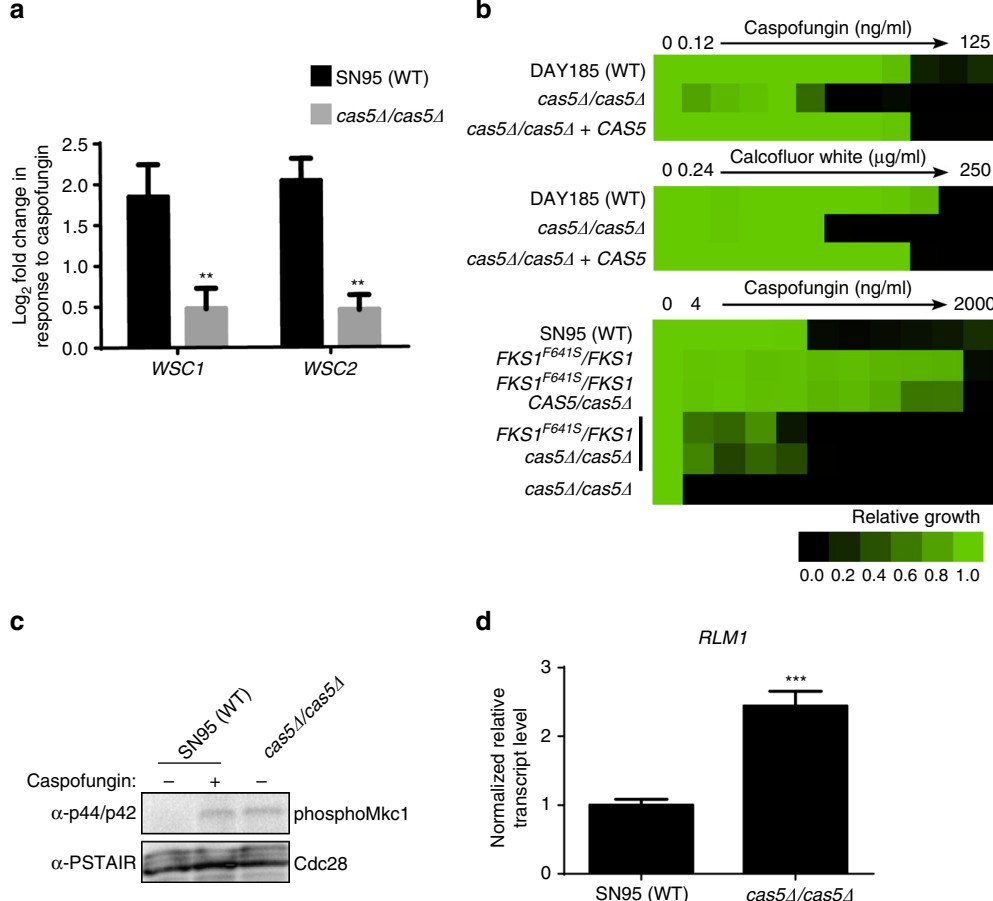

**Fig. 3** Cas5 regulates cell wall stability under basal and cell wall stress conditions. **a** *WSC1* and *WSC2*, transmembrane sensors that respond to cell wall perturbation, require Cas5 for upregulation in response to caspofungin. The transcript level of *WSC1* and *WSC2* were monitored by qRT-PCR and normalized to *TEF1*. Plotted are the $\log_2$ fold-changes in transcript levels upon caspofungin treatment relative to untreated conditions in both the wildtype (*black*) and *cas5Δ/cas5Δ* mutant (*gray*). *Error bars* represent standard deviation (s.d.) from the mean of triplicate samples. Significance was measured with an unpaired *t* test in GraphPad Prism (\*\**P* < 0.01). **b** Cas5 is required for tolerance and resistance to cell wall stress. Caspofungin or calcofluor white susceptibility assays were conducted in YPD medium. Growth was measured by absorbance at 600 nm after 48 h at 30 °C. Optical densities were averaged for duplicate measurements. Data was quantitatively displayed with color using Treeview (see *color bar*). **c** Deletion of *CAS5* leads to the activation of the cell wall integrity pathway in the absence of cell wall stress. A SN95 wild-type strain and a *cas5Δ/cas5Δ* mutant were left untreated (−) or treated for 1 h with 125 ng/ml of caspofungin (+), as indicated. Phosphorylated Mkc1 was monitored by western blot and detected with α-p44/42 antibody. Full blots are shown in Supplementary Fig. 2a. Cdc28 was detected with α-PSTAIRE antibody and used as a loading control. **d** *RLM1*, a transcription factor downstream of Mkc1 in the cell wall integrity pathway, is induced in a *cas5Δ/cas5Δ* mutant in the absence of cell wall tress. The transcript level of *RLM1* was monitored by qRT-PCR and normalized to *TEF1*. Differences in transcript level upon caspofungin treatment relative to untreated conditions are plotted. *Error bars* represent s.d. from the mean of triplicate samples. Significance was measured with an unpaired *t* test in GraphPad Prism (\*\*\**P* < 0.001)

(Fig. 4d and Supplementary Data 6). Importantly, these trends were less apparent in a mutant lacking Cas5 (Fig. 4d and Supplementary Data 6). Overall, our results highlight the important role of Cas5 in ensuring appropriate cell cycle dynamics under basal conditions and in response to cell wall stress.

**Cas5 is activated by dephosphorylation.** Given our findings that Cas5 governs cell cycle dynamics and cell wall stress responses, we next explored the mechanisms by which it is regulated. Initially, we monitored the subcellular localization of Cas5 by indirect immunofluorescence in a strain in which both alleles of *CAS5* were HA-tagged. The Cas5-HA protein was functional and sufficient to confer wild-type tolerance to caspofungin (Supplementary Fig. 3a). Under basal conditions, Cas5 appeared diffuse throughout the cytoplasm, however following treatment with caspofungin, Cas5 translocated to the nucleus (Fig. 5a). Owing to the moderate expression of Cas5 under its native

promoter, we employed a strain in which we could overexpress Cas5 to facilitate its localization in the absence of stress. Overexpression of HA-tagged Cas5 confirmed that Cas5 was mostly cytoplasmic, as observed when under its native promoter (Fig. 5b). Overexpression also enabled us to observe a sub-population of cells in which Cas5 was localized to the nucleus even in the absence of cell wall stress (Fig. 5b). Next, we assessed changes in Cas5 abundance in response to caspofungin. *CAS5* transcript and protein levels were induced in response to cell wall stress, as measured by qRT-PCR and western blot analysis (Fig. 5c, d and Supplementary Fig. 2b). Thus, in response to cell wall perturbation by caspofungin, Cas5 is induced and translocates to the nucleus.

When monitoring Cas5 protein levels, we observed that the Cas5 protein demonstrated a downward band shift when cells were exposed to cell wall antagonists, indicative of a post-translational modification (Fig. 5e and Supplementary Fig. 2c). To confirm this shift was a consequence of cell wall stress,

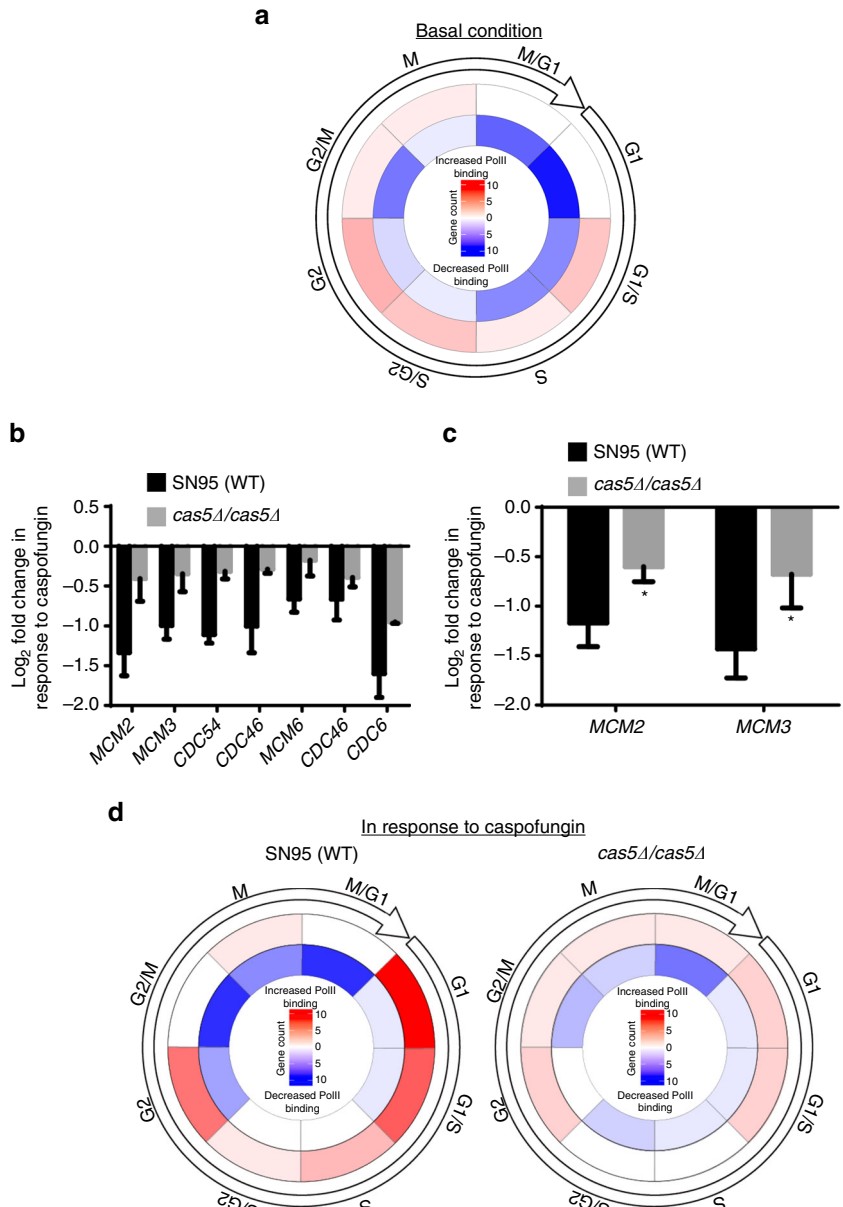

**Fig. 4** Cas5 governs cell cycle dynamics under basal and cell wall stress conditions. **a** Circos plot depicting the number of genes with increased (*red*) and decreased (*blue*) RNA PolII binding in the *cas5Δ/cas5Δ* mutant whose normal expression in the wild-type peaks at the indicated cell cycle stage. **b** Cas5 is required for the decreased PolII binding at genes encoding MCM complex members and *CDC6* in response to caspofungin. Histogram showing the PolII ChIP-Seq signal for the indicated genes encoding MCM complex components in response to caspofungin treatment. *Error bars* represent standard deviation (s.d.) from the mean of duplicate samples. **c** Downregulation of *MCM2* and *MCM3* in response to caspofungin depends on Cas5. Transcript levels were monitored by qRT-PCR and normalized to *TEF1*. *Error bars* represent s.d. from the mean of triplicate samples. Significance was measured with an unpaired *t* test in GraphPad Prism (*$P < 0.05$). **d** Circos plots illustrating the number of genes with increased (*red*) or decreased (*blue*) PolII binding in response to caspofungin (CF) in wild-type and *cas5Δ/cas5Δ* strains for genes whose expression is regulated throughout the cell cycle

we employed a strain in which we could specifically inhibit the kinase activity of the cell wall integrity regulator Pkc1[35]. In this strain, the only allele of *PKC1* carries an M850G gatekeeper mutation, rendering the kinase susceptible to inhibition by ATP analog 1-naphthyl-PP1 (1-NA-PP1), without affecting its kinase activity in the absence of the inhibitor[35, 36]. In the presence of 1-NA-PP1, the gatekeeper mutant was hypersensitive to caspofungin, similar to a *pkc1Δ/pkc1Δ* mutant (Supplementary Fig. 3b). Inhibition of Pkc1 kinase activity induced Cas5 protein levels and resulted in a downward mobility shift (Fig. 5f and Supplementary Fig. 2d), similar to what was observed in a wild-type strain treated with caspofungin (Fig. 5d). Given that a change in protein

phosphorylation is a common mechanism to regulate transcription factor activity[37], we assessed whether Cas5 was modified by phosphorylation. We performed two-dimensional gel electrophoresis coupled to western blot analysis. In the absence of stress, Cas5-HA resolved into a heterogeneous population of differentially charged species (Fig. 5g). In response to caspofungin, Cas5 collapsed into a less electronegative species, consistent with protein dephosphorylation. To confirm that Cas5 was phosphorylated under basal conditions, we monitored Cas5 mobility by western blot analysis following lambda phosphatase treatment. Phosphatase treatment led to a faster migrating Cas5 band, consistent with Cas5 being a phosphoprotein (Fig. 5h and

Supplementary Fig. 2e). Finally, to determine if dephosphorylation of Cas5 is coupled to the activation of gene expression, we monitored transcript levels of *ECM331* and *PGA13*, two Cas5-dependent caspofungin-responsive cell wall genes

(Supplementary Data 4)[22]. *ECM331* and *PGA13* were upregulated in wild-type cells with the same caspofungin exposure required to induce Cas5 dephosphorylation, and this upregulation was partially blocked by deletion of *CAS5* (Fig. 5i). Thus, Cas5 is

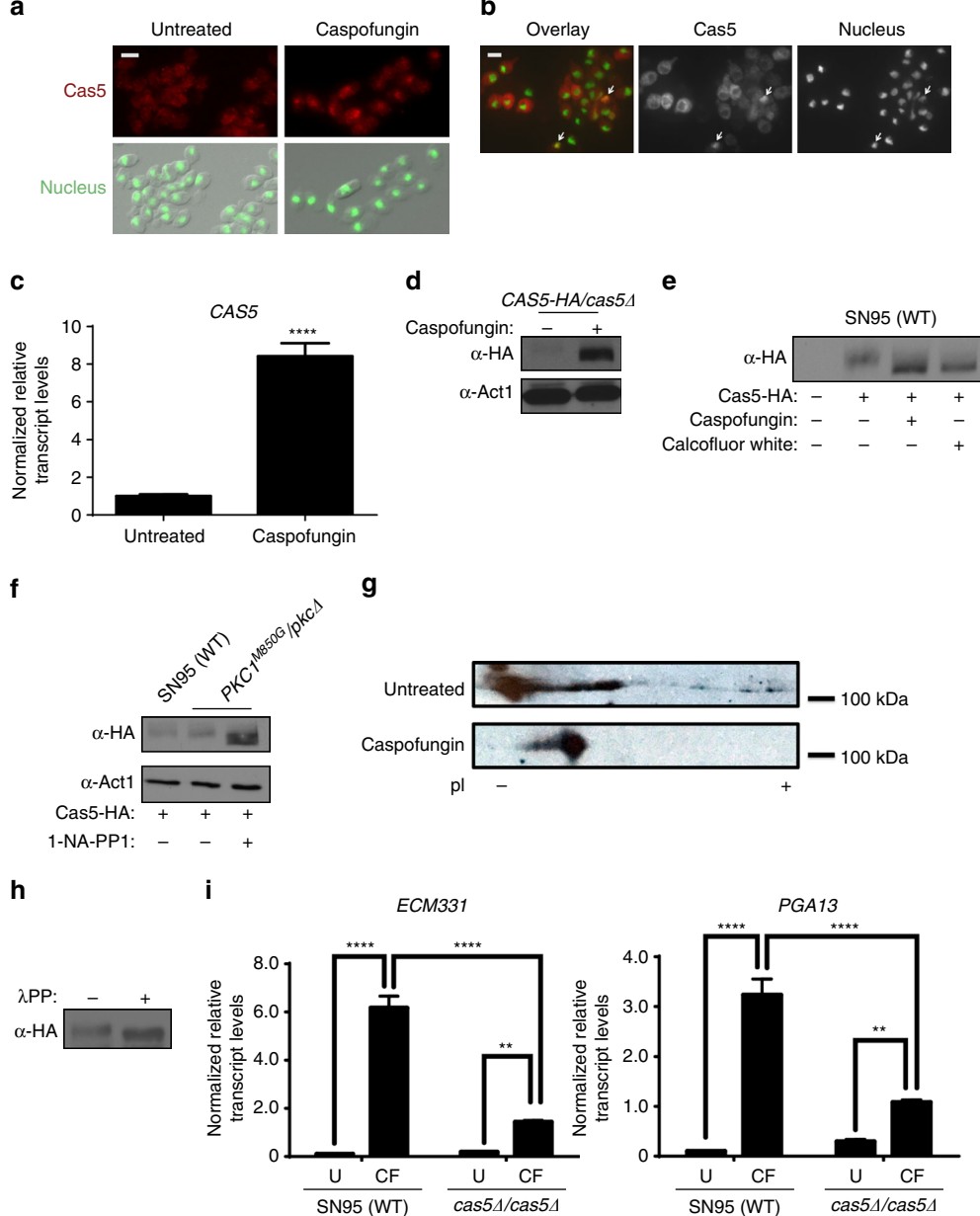

**Fig. 5** Cas5 becomes activated by dephosphorylation in response to cell wall stress. **a** Cas5 localizes to the nucleus in response to caspofungin. Cells were fixed and Cas5 (*red*) was detected by indirect immunofluorescence using α-HA antibody and α-mouse IgG-Cy3. Nuclei (*green*) were visualized by DAPI staining. *Scale bar* represents 5 μm. **b** Cells were subcultured in rich medium to log phase and fixed. HA-tagged Cas5 (*red*) and nuclei (*green*) were visualized as in **a**. *Scale bar* represents 5 μm. **c** *CAS5* is induced by caspofungin. The transcript level of *CAS5* was monitored by qRT-PCR and normalized to *GPD1*. *Error bars* represent standard deviation (s.d.) from the mean of triplicate samples. Significance was measured with an unpaired t test in GraphPad Prism (****$P < 0.0001$). **d** Levels of Cas5 were monitored by western blot and detected with an α-HA antibody. Actin was detected with an α-β-actin antibody as a loading control. Full blots are shown in Supplementary Fig. 2b. **e** Cas5 is post-translationally modified upon cell wall stress treatment. Cells were grown to log phase and subsequently treated with 125 ng/ml of caspofungin or 50 μg/ml of calcofluor white for 1 h. Total protein was resolved by SDS-PAGE and the blot was hybridized with an α-HA to monitor Cas5 migration. Full blots are shown in Supplementary Fig. 2c. **f** Cas5 migration and actin detection were monitored as part **d**. Full blots are shown in Supplementary Fig. 2d. **g** Protein lysates were subjected to two-dimensional gel electrophoresis coupled with western blotting. Cas5 was detected with an α-HA antibody. **h** Cas5 is phosphorylated in the absence of stress. Cas5 migration was monitored by western blot and detected with an α-HA antibody. Treatment of protein lysate with lambda phosphatase resulted in a faster migrating band. Full blots are shown in Supplementary Fig. 2e. **i** Cas5 is required for caspofungin (*CF*)-induction of cell wall genes. Transcript levels of *ECM331* and *PGA13* were monitored by qRT-PCR and normalized to *GPD1*. *Error bars* represent s.d. from the mean of triplicate samples. Significance was measured with a Tukey's multiple comparisons test in GraphPad Prism (****$P < 0.0001$, **$P < 0.01$)

regulated by dephosphorylation and governs the expression of caspofungin-responsive cell wall genes in response to cell wall stress.

**A Cas5 serine residue is critical for caspofungin tolerance.** On the basis of our observations that Cas5 function is regulated by phosphorylation, we hypothesized that dephosphorylation of key residues is important for Cas5 activation. To map the phosphorylation residues on Cas5 important for its activity, we performed immunoprecipitation coupled to mass spectrometry with HA-tagged Cas5. Phosphorylation was detected at serine residues S462 and S476 (Supplementary Fig. 4a). To test whether these serine residues along with two other adjacent residues, S464 and S472, were functionally important for Cas5 function in regulating cell wall integrity, we mutagenized all four serines to either glutamic acid, to mimic a constitutively phosphorylated state, or alanine, to mimic a constitutively unphosphorylated state. The resulting mutants displayed wild-type tolerance to caspofungin (Supplementary Fig. 4b), with no appreciable difference in the band shift upon caspofungin treatment (Supplementary Fig. 4c), suggesting that the sites identified by mass spectrometry were not sufficient to control Cas5 function.

To identify other potential Cas5 phosphorylation sites, we analyzed its amino acid sequence using NetPhos and followed up on S769, a highly conserved residue in one of the zinc finger domains with a phosphorylation site-prediction score of 0.980 (Fig. 6a)[38, 39]. To assess the functional significance of this residue, we engineered strains in which one allele of *CAS5* was deleted and the other was replaced with a wild-type allele, the phosphomimetic S769E allele, or the phosphodeficient S769A allele. Introduction of the phosphomimetic S769E allele as the only source of *CAS5* phenocopied homozygous deletion of *CAS5* in terms of hypersensitivity to caspofungin, whereas introduction of the phosphodeficient S769A allele conferred a wild-type phenotype (Fig. 6b). Consistent with our hypothesis, the S769E mutation blocked the upregulation of *ECM331* and *PGA13* in response to caspofungin, whereas the S769A mutation did not (Fig. 6c). Notably, these mutations did not affect Cas5 induction or dephosphorylation in response to caspofungin (Fig. 6d and Supplementary Fig. 2f), suggesting that the S769E substitution does not affect the capacity of Cas5 to sense cell wall stress, and that additional phosphorylation sites remain to be identified. Our data supports a model in which Cas5 is activated by dephosphorylation in response to cell wall stress.

**Glc7 dephosphorylates Cas5 in response to cell wall stress.** To identify the phosphatase that regulates Cas5 activity, we leveraged insights from the model yeast *S. cerevisiae*. A BLASTp search of the *C. albicans* Cas5 protein sequence against the *S. cerevisiae* protein database identified Mig1, Msn4, Mig2, and Msn2 as proteins with similar sequences[40]. Mig1, Msn4, and Msn2 are dephosphorylated by the type I protein phosphatase Glc7[41, 42]. As Glc7 influences cell wall maintenance in *S. cerevisiae*[43], we hypothesized that Glc7 was required for activating cell wall stress responses in *C. albicans*. We engineered a strain in which we could transcriptionally repress *GLC7* by deleting one allele and replacing the native promoter of the remaining allele with a tetracycline-repressible promoter. We confirmed that *GLC7* expression was repressed in the presence of the tetracycline analog doxycycline by qRT-PCR (Supplementary Fig. 5a). Depletion of *GLC7* with doxycycline conferred hypersensitivity to caspofungin (Fig. 7a), confirming Glc7 is important for tolerating cell wall stress in *C. albicans*. Next, we examined the effect of *GLC7* depletion on Cas5 expression and function. Doxycycline-mediated transcriptional repression of *GLC7* did not block

upregulation of Cas5 in response to caspofungin (Fig. 7b and Supplementary Fig. 2g), but did block the downward band shift associated with dephosphorylation, as was evident with equivalent Cas5 levels loaded per sample (Fig. 7c and Supplementary Fig. 2h). Depletion of *GLC7* also blocked the upregulation of Cas5-dependent caspofungin-responsive genes, *ECM331* and *PGA13* (Fig. 7d). Thus, Glc7 activates Cas5 by dephosphorylation in response to cell wall stress, thereby enabling transcriptional upregulation of cell wall genes.

**Cas5 governs cell wall stress responses with Swi4/Swi6.** To identify physical interactors of Cas5, we leveraged the results from our immunoprecipitation coupled to mass spectrometry analysis (Supplementary Data 7). The only interaction that passed a stringent statistical analysis using Significance Analysis of Interactome (SAINT)[44] with a probability threshold of 0.9 was Swi4 (Supplementary Data 7), which interacts with Swi6 to form the SBF complex in *S. cerevisiae*[45]. To validate the interaction between Cas5 and the SBF complex, we individually tandem affinity purification (TAP) tagged Swi4 or Swi6 at the C terminus in a strain harboring Cas5-HA to enable co-immunoprecipitation analysis. The Swi4-TAP and Swi6-TAP proteins were functional and sufficient to mediate wild-type tolerance to caspofungin (Supplementary Fig. 5b). Immunoprecipitation of Cas5 with anti-HA agarose co-purified both Swi4 and Swi6 (Fig. 8a and Supplementary Fig. 6), verifying these proteins physically interact. To determine whether the SBF complex has a role in cell wall stress responses in *C. albicans*, we monitored the expression of *ECM331* and *PGA13* in mutants lacking Swi4, Swi6, or Cas5. In all three mutants, upregulation of these cell wall genes in response to caspofungin was blocked (Fig. 8b). However, strains lacking Swi4 or Swi6 displayed intermediate caspofungin susceptibility phenotypes relative to a wild-type strain and a *CAS5* homozygous deletion mutant (Fig. 8c). Altogether, our results suggest that Cas5 regulates transcriptional responses to cell wall stress in part through the SBF complex and in part through an independent mechanism.

**Cas5 governs cell cycle dynamics independent of Swi4/Swi6.** Since our RNA PolII ChIP-Seq data implicated Cas5 in proper cell cycle dynamics and DNA replication (Figs. 1c and 4), we wanted to further investigate the impact of Cas5 on nuclear division. We monitored the number of nuclei per cell in wildtype and *cas5Δ/cas5Δ* strains. Cells were stained with DAPI for DNA content and calcofluor white for chitin (Fig. 9a and Supplementary Fig. 7a). During log phase, ~40% of cells lacking Cas5 were multinucleated, an indication of uncontrolled DNA replication or a defect in cytokinesis (Fig. 9b). Given that Swi4 and Swi6 physically interact with Cas5 (Fig. 8a) and that the SBF complex regulates the G1/S phase progression in *S. cerevisiae* and *C. albicans*[46, 47], we also investigated their impact on nuclear division. Strikingly, <10% of mutant cells lacking either Swi4 or Swi6 showed a multinucleated phenotype, suggesting that the SBF complex has a minor role in ensuring proper nuclear division in *C. albicans* (Fig. 9a, b). We further investigated the defective nuclear division associated with loss of Cas5 by time-lapsed fluorescence microscopy. To monitor DNA content, we RFP-tagged histone H4, encoded by *HHF1*; to monitor mitotic spindle positioning, we GFP-tagged the Dam1 complex subunit *DAD2* (Fig. 9c, d). We observed multiple nuclei dividing simultaneously in a single cell in a *cas5Δ/cas5Δ* mutant (Fig. 9d and Supplementary Fig. 7b). Importantly, each nucleus had at least one spindle pole body, suggesting that the accumulation of nuclei was due to nuclear division and not spontaneous nuclear fragmentation (Fig. 9d and Supplementary Fig. 7b).

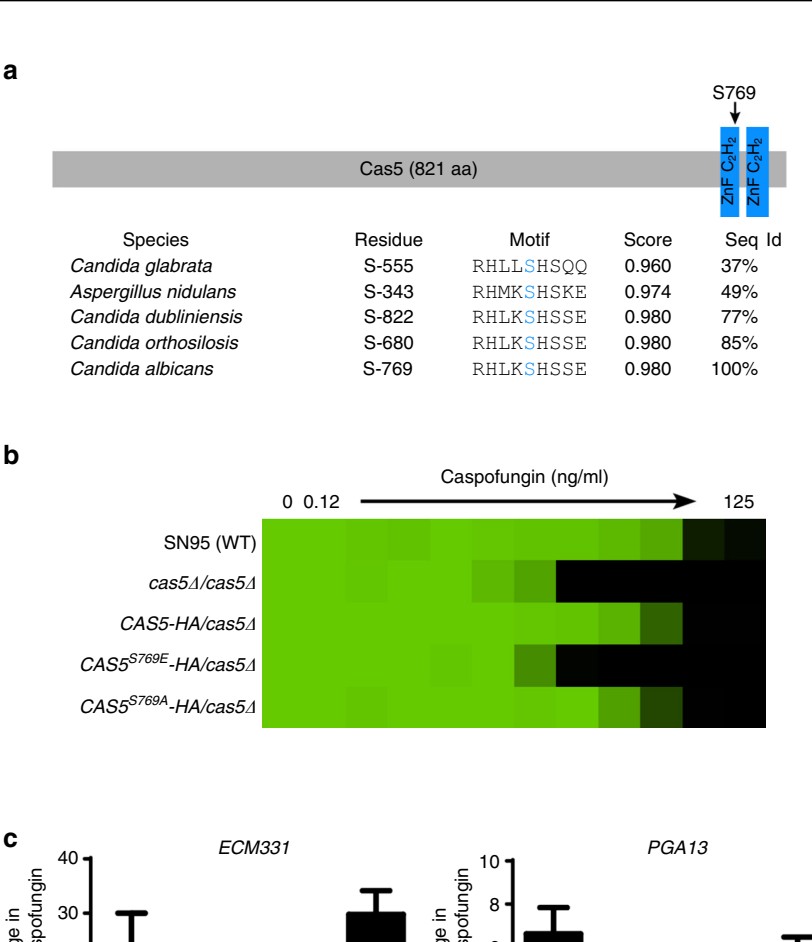

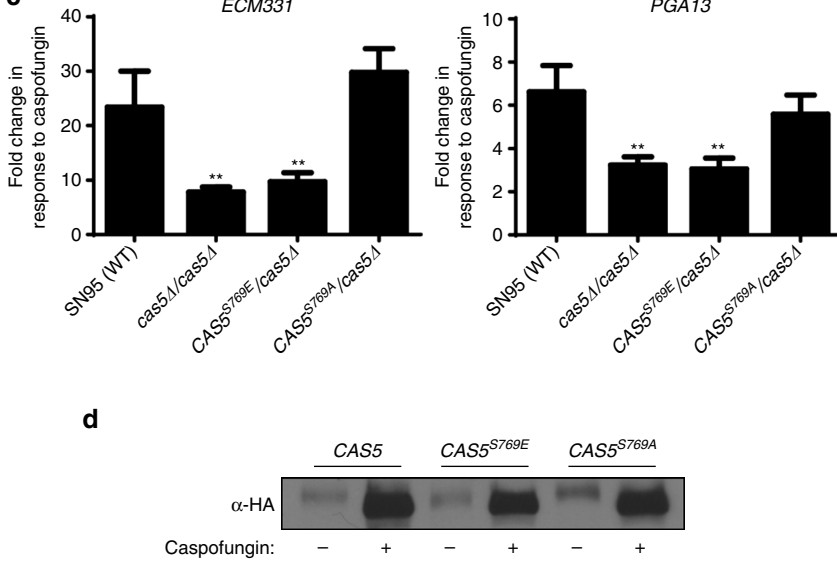

**Fig. 6** The S769E substitution in the Cas5 DNA-binding domain phenocopies *CAS5* deletion. **a** Schematic showing the position of S769 in the Cas5 zinc finger domain and the alignment of Cas5 orthologs in related fungal species. The alignment was performed using the *Candida* Genome Database (CGD), and the phosphorylation site-prediction score was generated using NetPhos. **b** The phosphomimetic S769E substitution in Cas5 confers hypersensitivity to caspofungin. Caspofungin susceptibility assays were conducted in YPD medium. Growth was measured by absorbance at 600 nm after 48 h at 30 °C. Optical densities were averaged for duplicate measurements. Data was quantitatively displayed with color using Treeview (see *color bar* in Fig. 2). **c** The S769E substitution blocks induction of cell wall genes in response to caspofungin treatment. The transcript levels of *ECM331* and *PGA13* were monitored by qRT-PCR and normalized to *GPD1*. Plotted are the fold-changes in transcript levels of *ECM331* or *PGA13* in response to caspofungin relative to untreated. *Error bars* represent standard deviation (s.d.) from the mean of triplicate samples. The fold change in gene expression for each mutant strain was compared with the wild-type strain using one-way ANOVA in GraphPad Prism (**$P < 0.01$). **d** Phosphomutations in *CAS5* do not affect band shifts associated with activation of the cell wall stress response, as observed upon caspofungin treatment. Cas5 was monitored by western blot and detected using an α-HA antibody. Full blots are shown in Supplementary Fig. 2f

Next, we assessed the ploidy of the *cas5Δ/cas5Δ* mutant compared with a wild-type control by flow cytometry. During log phase growth in the absence of external stress, wild-type cells exhibited two distinct peaks at 2C and 4C (representing the G1- and G2-phases of the cell cycle, respectively), whereas the *cas5Δ/cas5Δ* mutant exhibited five distinct peaks at 2C, 4C, 6C, 8C, and 12C (Fig. 9e and Supplementary Fig. 7c). Notably, peaks at 1C and 3C were absent, indicating that the nuclei were not haploidizing, mating, or undergoing reductional division. Instead, these data suggest multiple rounds of DNA replication were

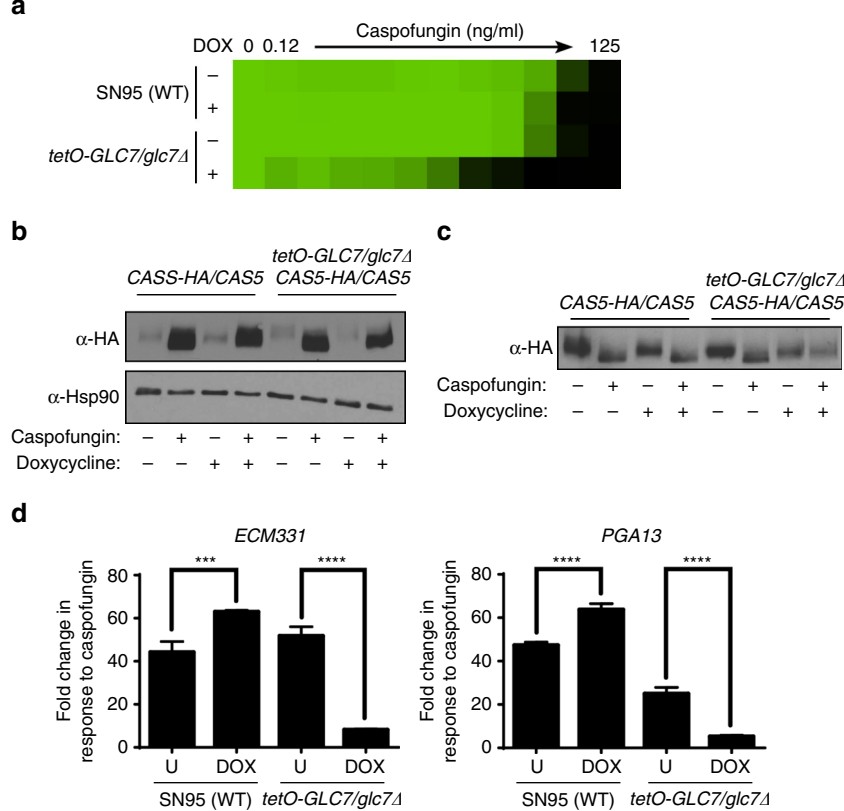

**Fig. 7** Cas5 activation in response to caspofungin is coupled with dephosphorylation by the protein phosphatase Glc7. **a** Depletion of *GLC7* confers sensitivity to caspofungin. Caspofungin susceptibility assays were conducted in YPD medium in the presence or absence of 1 µg/ml of doxycycline (DOX). Growth was measured by absorbance at 600 nm after 48 h at 30 °C. Optical densities were averaged for duplicate measurements. Data were quantitatively displayed with *color* using Treeview (see *color bar* in Fig. 2). **b** Upregulation of Cas5 expression does not depend on Glc7. *CAS5-HA/CAS5* and *CAS5-HA/CAS5 tetO-GLC7/glc7Δ* strains were cultured in the absence or presence of doxycycline and caspofungin, as indicated. Cas5 was monitored by western blot and detected with an α-HA antibody. Hsp90 protein levels served as a loading control. Full blots are shown in Supplementary Fig. 2g. **c** Post-translational modification of Cas5 is absent upon *GLC7* depletion. The western blot was performed as described in **b**, except caspofungin treated samples were diluted fivefold to achieve equal loading of Cas5. Full blots are shown in Supplementary Fig. 2h. **d** Dephosphorylation of Cas5 is required for the induction of cell wall genes in response to caspofungin treatment. *CAS5-HA/CAS5* and *CAS5-HA/CAS5 tetO-GLC7/glc7Δ* strains were grown in the absence or presence of doxycycline (*DOX*) or caspofungin, as indicated. The transcript levels of *ECM331* and *PGA13* were monitored by qRT-PCR and normalized to *GPD1*. Plotted are the fold-changes in transcript levels in the presence of caspofungin relative to untreated. *Error bars* represent standard deviation (s.d.) from the mean of triplicate samples. The treatment conditions were compared using a Tukey's multiple comparisons test in GraphPad Prism (****$P < 0.0001$, ***$P < 0.001$).

occurring in the same cell, causing ploidy level increases similar to what is observed when *C. albicans* is exposed to fluconazole[48].

Finally, to determine whether the dephosphorylated form of Cas5 was required for controlled DNA replication, we quantified the number of nuclei per cell in the phosphomutants. The *CAS5* mutant carrying the phosphomimetic S769E substitution phenocopied the cell cycle defects observed in the *CAS5* homozygous deletion mutant, whereas the mutant carrying phosphodeficient S769A substitution was indistinguishable from the wildtype (Fig. 9f). Thus, our results support a novel role for the dephosphorylated form of Cas5 in regulating nuclear division that is largely independent of the SBF complex.

## Discussion

The capacity to coordinate cell cycle progression with stress responses is crucial for cellular survival to give cells time to recover from a myriad of environmental perturbations, including cell wall stress[49]. In the present work, we uncovered a dual role for the transcription factor Cas5 in mediating cell cycle dynamics and cell wall stress responses in *C. albicans* (Fig. 10). We

discovered that RNA PolII binding to many genes important for cell wall biosynthesis and DNA replication is dependent on Cas5 under basal conditions (Fig. 1). We also found Cas5 regulates the expression of a distinct set of cell wall and cell cycle genes in response to cell wall stress (Fig. 2). We established Cas5 function is regulated by the protein phosphatase Glc7 (Fig. 7), and Cas5 functions in concert with Swi4 and Swi6 to regulate cell wall homeostasis (Fig. 8). Our findings implicate Cas5 at the core of a novel mechanism by which cell cycle and cell wall integrity are coordinately regulated.

The coordination of sensing and responding to cell wall stress is crucial for fungi, and precise temporal control of cell wall stress response regulators underpins the rapid mobilization of signaling cascades that govern cellular integrity. We demonstrated that Cas5 not only becomes activated in response to cell wall stress but it also maintains cell wall stability under basal conditions (Figs. 3 and 5). Although Cas5 is not broadly conserved in the fungal kingdom, its activation is regulated by a conserved protein phosphatase, Glc7, which has 91% protein identity to the *S. cerevisiae* ortholog. In *S. cerevisiae*, temperature sensitive *glc7* mutants exhibit sorbitol-remediable lysis defects at the restrictive

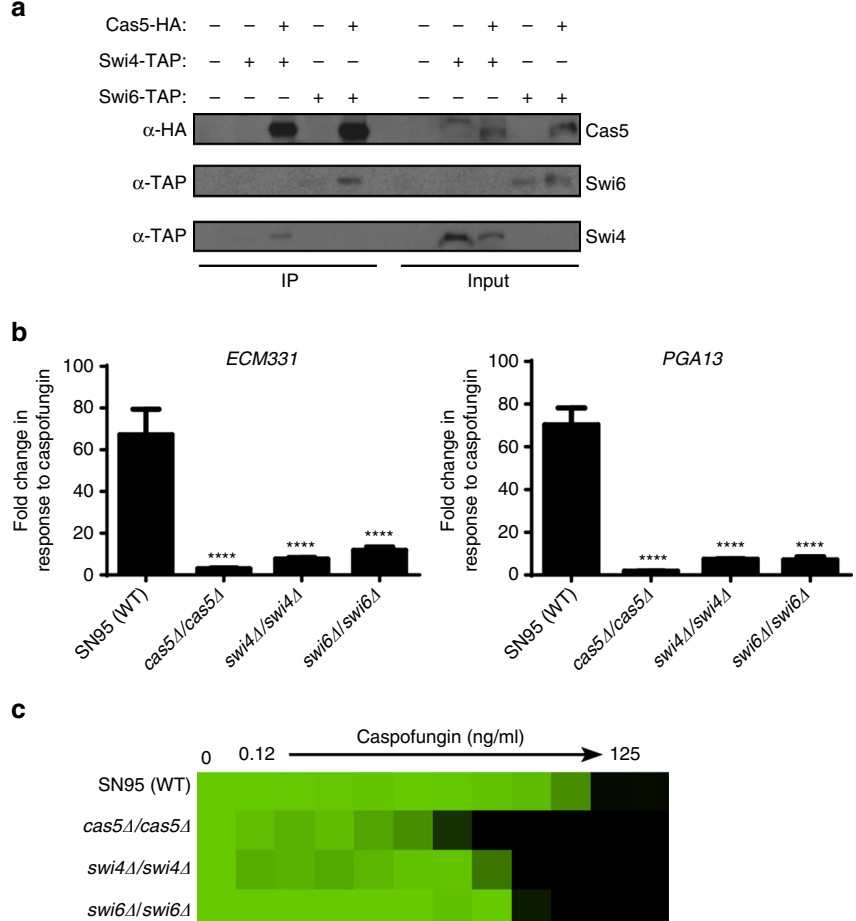

**Fig. 8** Cas5 regulates caspofungin tolerance in part through the interaction with components of the SBF complex, Swi4 and Swi6. **a** Swi4 and Swi6 co-purify with Cas5. C terminally HA-tagged Cas5 was immunoprecipitated with α-HA beads. Swi4 and Swi6 co-purification was monitored by western blot and detected with an α-TAP antibody. Cas5 pull down was confirmed by detection with an α-HA antibody. Input samples confirm the expression of tagged proteins. Full blots are shown in Supplementary Fig. 6. **b** Cas5, Swi4, and Swi6 are required for the upregulation of caspofungin-responsive genes. The transcript levels of *ECM331* and *PGA13* was monitored by qRT-PCR and normalized to *GPD1*. Plotted are the fold-changes in transcript levels in the presence of caspofungin relative to untreated. *Error bars* represent standard deviation (s.d.) from the mean of triplicate samples. The fold change in gene expression for each mutant strain was compared to the wildtype using one-way ANOVA in GraphPad Prism (****$P < 0.0001$). **c** Swi4 and Swi6 regulate cell wall stress response. Caspofungin susceptibility assays were conducted in YPD medium. Growth was measured by absorbance at 600 nm after 48 h at 30 °C. Optical densities were averaged for duplicate measurements. Data was quantitatively displayed with color using Treeview (see *color bar* in Fig. 2)

temperature, due to impaired cell wall integrity[43]. We demonstrated Glc7 is required for cell wall maintenance in *C. albicans* (Fig. 7). Glc7 substrates in *S. cerevisiae* include three proteins with sequence similarity to Cas5: Mig1, Msn2, and Msn4. Msn2 and Msn4 are transcription factors that are required for general stress response in *S. cerevisiae*, but not in *C. albicans*[50, 51]. This highlights a divergence in signal transduction cascades required to respond to environmental stressors, and suggests that Cas5 may be the central downstream effector of Glc7 that modulates stress responses in *C. albicans*. Consistent with this possibility, Cas5 controls responses not only to cell wall stress, but also to cell membrane stress exerted by the azole antifungal drugs[52].

In yeast, cell wall stress originates not only through environmental perturbations but also through normal physiological changes that are coupled to cell cycle progression, as cell wall remodeling and biosynthesis are required to enable the emergence of the growing daughter bud. In *S. cerevisiae* cell membrane perturbation triggers cell cycle arrest via degradation of Cdc6, a component of the pre-RC[53]. This is required for cell survival as continued cell cycle progression in the presence of plasma membrane damage induces plasma membrane rupture and cell

lysis, leading to cell death[53]. We found that *CDC6* and genes encoding members of the MCM complex that participate in the pre-RC are transcriptionally repressed in response to caspofungin in a Cas5-dependent manner (Fig. 4). To our knowledge, this is the first mechanistic insight into the signaling events that couple cell wall stress with cell cycle arrest in *C. albicans*, and likely enables the cell to respond to environmental stress without succumbing to cell death. In *S. cerevisiae*, cell cycle genes are regulated by transcriptional complexes composed of Swi4 and Swi6, including the SBF complex, which is involved in budding and membrane/cell-wall biosynthesis[54]. We observed that impairment of Cas5 function manifested in more severe misregulation of nuclear division than deletion of *SWI4* or *SWI6* (Fig. 9), suggesting Cas5 regulates nuclear division largely independent of the SBF complex.

Targeting regulators of cellular circuitry that are crucial for cellular stress responses may provide a powerful strategy for antifungal drug development. This strategy opens up the opportunity for combining stress response inhibitors with current antifungals for treatment of fungal infections. Promising examples of this approach include targeting the molecular chaperone

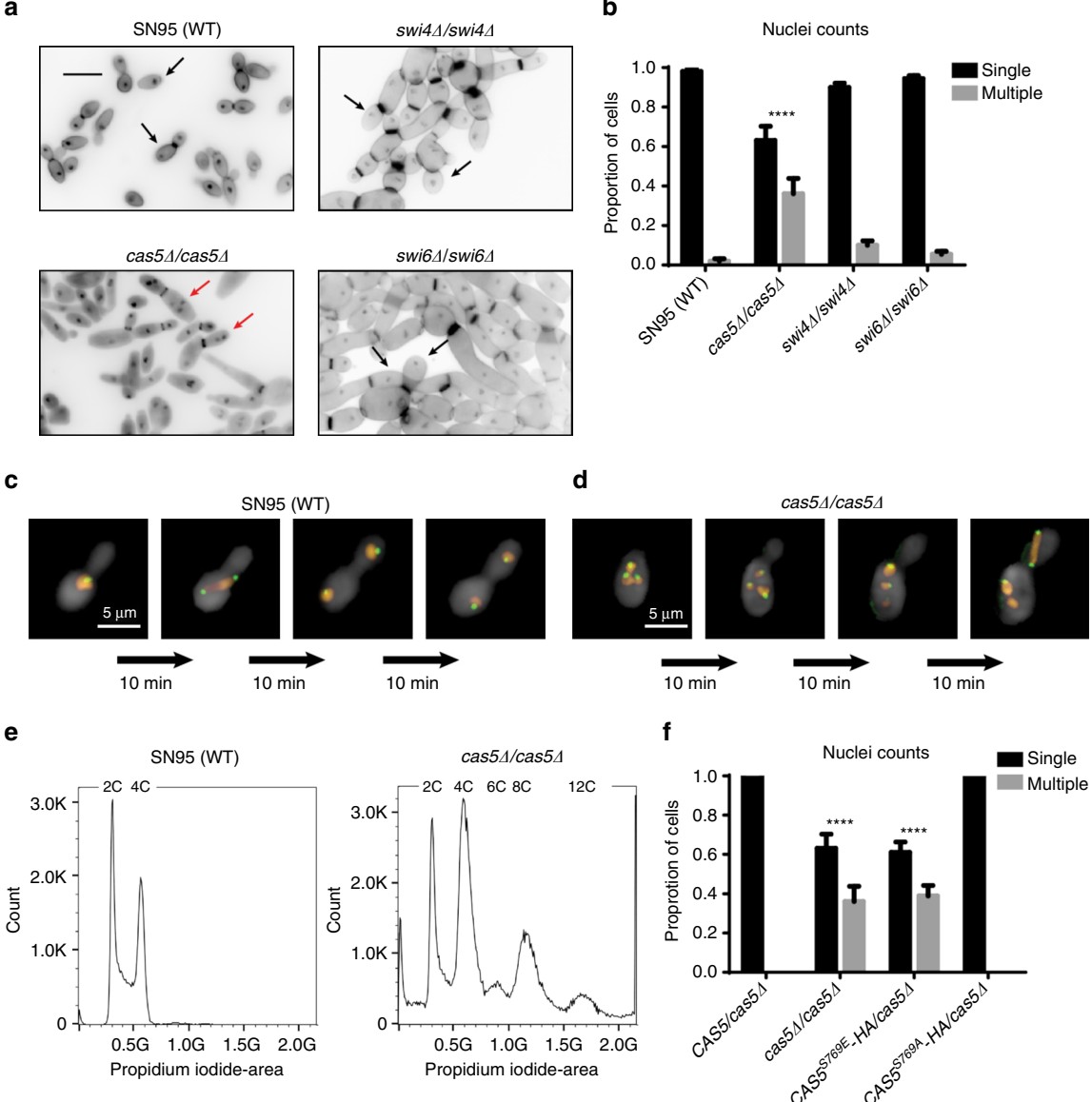

**Fig. 9** Cas5 regulates nuclear division largely independent of Swi4 and Swi6. **a** Nuclei were stained with DAPI and chitin was stained using calcofluor white. For each image, 12 Z-stack slices were taken at 0.3 μm each. *Black arrows* highlight cells with a single nucleus. *Red arrows* highlight cells with a multinucleated phenotype. **b** Mutants lacking Cas5 exhibit severe cell cycle defect. The histogram represents the average number of nuclei per cell in three biological replicates. The number of nuclei counted in each experiment was at least 120 cells for each strain. The nuclei counts for each strain were averaged and *error bars* represent standard deviation (s.d.) from the mean of biological duplicates. Nuclei counts for each strain were compared to the wildtype using one-way ANOVA in GraphPad Prism (****$P < 0.0001$). **c, d** Mitotic spindles are aligned along the mother–bud axis during cell division in wild-type cells (**c**) and misaligned in a mutant lacking Cas5 (**d**). DNA was monitored by RFP-tagged Hhf1, and mitotic spindle was monitored by GFP-tagged Dad2 using time-lapse fluorescence microscopy. **e** Cas5 is required for maintaining normal DNA content throughout the cell cycle. Cellular DNA content was measured by propidium iodide and flow cytometry of the wild-type diploid and the *cas5Δ/cas5Δ* mutant strain. The wild-type diploid has the standard G1 and G2 cell cycle peaks representing 2C and 4C DNA levels. The *cas5Δ/cas5Δ* mutant population, in addition to the same 2C and 4C DNA levels, contained a large subpopulation of cells with DNA levels at 6C, 8C, and 12C. These DNA levels represent tetraploid (4N) and hexaploid (6N) cells. **f** Activation of Cas5 by dephosphorylation is required for proper cell cycle progression. The histogram represents the average number of nuclei per cell in three biological replicates. The number of nuclei counted was at least 120 cells for each strain. The nuclei counts for each strain were averaged and *error bars* represent s.d. from the mean of biological duplicates. Nuclei counts for each strain were compared to that of the wildtype using one-way ANOVA in GraphPad Prism (****$P < 0.0001$)

Hsp90 or its downstream effector calcineurin, which are both required for drug resistance and virulence of diverse fungal pathogens[8, 55]. Given that both of these regulators are highly conserved in humans and have essential functions in mobilizing immune responses, the challenge in exploiting the therapeutic utility of these targets for antifungal therapies hinges on the development of fungal-selective inhibitors[56]. Our work further

supported the role of Cas5 in governing echinocandin tolerance and established that Cas5 enables echinocandin resistance in a strain harboring a mutation in the drug target gene *FKS1* (Fig. 3b). As Cas5 lacks an identifiable ortholog in humans[22], but is required for drug resistance (Fig. 3b) and virulence in *C. albicans*[10, 57], it provides an attractive target for antifungal drug development. There is growing support for the feasibility of

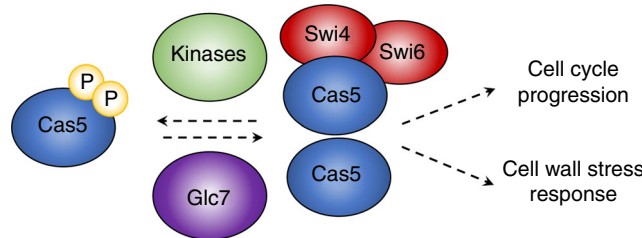

**Fig. 10** Model for Cas5-mediated coupling of cell cycle progression to cell wall stress response. Cas5 is dephosphorylated by the protein phosphatase Glc7 in response to cell wall stress. Once dephosphorylated, it can translocate to the nucleus to govern the expression of a myriad of genes, including those important for cell cycle regulation and cell wall stress response. Cas5 regulates changes in gene expression through both Swi4/Swi6-dependent and independent mechanisms

chemical modulation of transcription factors based on blocking transcription factor dimerization, DNA binding, or the interaction with regulatory proteins[58, 59]. Elucidating mechanisms that enable remodeling of transcriptional programs that control stress response and virulence traits provides opportunities to expand the antimicrobial target space in this era of antimicrobial resistance.

## Methods

**Strains and reagents**. All *C. albicans* strains were archived in 25% glycerol and stored at −80 °C. Strains were grown in YPD (1% yeast extract, 2% bactopeptone, and 2% glucose) at 30 °C unless otherwise specified. For solid media, 2% agar was added. Strains were constructed according to standard protocols. Strains used in this study are listed in Supplementary Table 1. Caspofungin was generously provided by Terry Roemer (Department of Infectious Diseases, Merck Research Laboratories) and was diluted to a 100 µg/ml stock, and used at a final concentration of 125 ng/ml. Doxycycline (DOX, BD Biosciences #631311) was formulated in a 20 mg/ml stock in water and used at a final concentration of 0.02 µg/ml or 1 µg/ml, as specified.

**Strain construction**. CaLC2034: To generate a *CAS5* heterozygous deletion mutant, the NAT flipper cassette (pLC49)[60] was PCR amplified using primers oLC2017 and oLC2018 (4366 bp) and transformed into CaLC239 (SN95). NAT-resistant transformants were PCR tested with oLC275 in combination with oLC2034 for upstream integration and oLC274 in combination with oLC2035 for downstream integration. The *SAP2* promoter was induced to drive expression of FLP recombinase to excise the NAT flipper cassette.

CaLC2056: To generate a *CAS5* homozygous deletion mutant, the NAT flipper cassette (pLC49)[60] was PCR amplified as described for CaLC2034 and transformed into CaLC2034. NAT-resistant transformants were PCR tested for correct integration of the cassette as described for CaLC2034. The presence of deleted allele was verified by amplification with primer pairs oLC2034 in combination with oLC2035, and absence of wild-type allele was verified by amplification with primer pairs oLC2047 in combination with oLC2048. The *SAP2* promoter was induced to drive expression of FLP recombinase to excise the NAT flipper cassette.

CaLC3908/CaLC3909: To generate a *CAS5* heterozygous deletion mutant in a strain carrying the *FKS1*[F641S] mutation, the NAT flipper cassette (pLC49)[60] was PCR amplified as described for CaLC2034 and transformed into CaLC2087 (SN95 *FKS1*[F641S]/*FKS1*). Correct upstream and downstream integration at the *CAS5* locus was verified as described for CaLC2034. The *SAP2* promoter was induced to drive expression of FLP recombinase to excise the NAT flipper cassette. To generate a *CAS5* homozygous deletion mutant, the NAT flipper cassette (pLC49)[60] was again PCR amplified and transformed into the SN95 *FKS1*[F641S]/*FKS1 CAS5/cas5::FRT* strain. Correct upstream and downstream integration at the *CAS5* locus was verified as described for CaLC2034. The absence of a wild-type allele was confirmed with oLC2034 in combination with oLC2164. The *SAP2* promoter was induced to drive expression of FLP recombinase to excise the NAT flipper cassette.

CaLC3378: To C terminally HA tag *CAS5* in a mutant carrying the *PKC1*[M850G] mutation, the *CAS5*-HA-HIS cassette was released from pLC790 with *Apa*I and transformed into SN95 *PKC1*[M850G]/*PKC1*. Proper integration at the *CAS5* locus was verified by amplification using primer pairs oLC2163 in combination with oLC2029 and oLC2164 in combination with oLC1645. Expression of HA-tagged Cas5 was verified by western blot.

CaLC3859: To regulate the expression of *CAS5*, the tetracycline-repressible transactivator, the *tetO* promoter, and the NAT flipper cassette were PCR amplified from pLC605[61] using primers oLC2088 in combination with oLC2089. The PCR amplified product was transformed into CaLC2034. Correct upstream and

downstream integration at the *CAS5* locus was verified by amplifying across both junctions using primer pairs oLC2034 in combination with oLC534 and oLC300 in combination with oLC2145. The absence of a wildtype *CAS5* promoter was verified with oLC2034 in combination with oLC2035. To C terminally HA tag *CAS5*, the *CAS5*-HA-*HIS1* cassette was released from pLC818 with *Sac*II and transformed into the *tetO-CAS5/cas5*Δ. Correct upstream integration at the C terminus of *CAS5* was verified by amplifying across the junction using primer pair oLC2163 in combination with oLC2164. Expression of HA-tagged Cas5 was verified by western blot.

CaLC3113: To C terminally HA tag *CAS5* in a mutant heterozygous for *CAS5*, the HA-*HIS1* cassette was PCR amplified from pLC575[62] using primers oLC2161 in combination with oLC2162 and transformed into CaLC2034. Transformants prototrophic for histidine were PCR tested with C2163 in combination with oLC2029 for upstream integration and oLC2164 in combination with oLC1645 for downstream integration.

CaLC3151: To C terminally HA tag *CAS5* in a mutant heterozygous for *CAS5*, the *CAS5*-HA-*HIS1* cassette was released from pLC790 with *Apa*I and transformed into CaLC2034. Transformants prototrophic for histidine were PCR tested with oLC2163 in combination with oLC2029 for upstream integration and oLC2164 in combination with oLC1645 for downstream integration. This strain shares the same genotype as CaLC3113 but was generated as a control for strains made using plasmid pLC790 as the backbone, such pLC791 and pLC800.

CaLC4285: To C terminally HA tag *CAS5*, the HA-*ARG4* cassette was PCR amplified from pLC576[62] using primers oLC2161 in combination with oLC2162 and transformed into CaLC1900. Correct integration at the C terminus of *CAS5* was verified by using primer pair oLC2163 in combination with oLC2164.

CaLC2213: To C terminally HA tag *CAS5* in SN95, the HA-*HIS1* cassette was PCR amplified from pLC575[62] using primers oLC2161 in combination with oLC2162 and transformed into CaLC239 (SN95). Transformants prototrophic for histidine were PCR tested for correct integration of the cassette with oLC2163 in combination with oLC2029 and oLC2164 in combination with oLC1645.

CaLC3044: To C terminally HA tag the both alleles of *CAS5*, the HA-*ARG4* cassette was PCR amplified from pLC576[62] as described for CaLC4285 and transformed into CaLC2213. Transformants prototrophic for both histidine and arginine were PCR tested for correct integration of the cassette as described for CaLC4285.

CaLC3209: To introduce a mutant allele of *CAS5* carrying the S769E mutation, the *CAS5*[S769E]-HA-*HIS1* cassette was released from pLC791 with *Apa*I and transformed into CaLC2034. Transformants prototrophic for histidine were PCR tested for integration of the cassette with oLC3052 in combination with oLC2029 and oLC2164 in combination with oLC1645. The absence of a wildtype *CAS5* allele was verified with oLC3052 in combination with oLC2164. The S769E mutation was sequence verified with oLC2029.

CaLC3189: To introduce a mutant allele of *CAS5* carrying the S769A mutation, the *CAS5*[S769A]-HA-*HIS1*cassette was released from pLC800 with *Apa*I and transformed into CaLC2034. Transformants prototrophic for histidine were PCR tested for integration of the cassette as described for CaLC3209. The S769A mutation was sequence verified with oLC2029.

CaLC4036: To generate a *SWI4* homozygous deletion mutant, the NAT flipper cassette (pLC49)[60] was PCR amplified using primers oLC3472 in combination with oLC3473 and transformed into CaLC239 (SN95). NAT-resistant transformants were PCR tested with oLC275 in combination with oLC3474 and oLC274 in combination with oLC3427. The *SAP2* promoter was induced to drive expression of FLP recombinase to excise the NAT flipper cassette. Once again, the NAT flipper cassette (pLC49)[60] was PCR amplified using primers oLC3472 in combination with oLC3473 and transformed into *SWI4/swi4::FRT*. NAT-resistant transformants were PCR tested as described above. The absence of a wild-type allele was verified with oLC3426 in combination with oLC3427, and the presence of deleted allele was verified by oLC3473 in combination with oLC3427. The *SAP2* promoter was induced to drive expression of FLP recombinase to excise the NAT flipper cassette.

CaLC4330: To generate a *SWI6* homozygous deletion mutant, the NAT flipper cassette (pLC49)[60] was PCR amplified using primers oLC3641 in combination with oLC3642 and transformed into CaLC239 (SN95). NAT-resistant transformants were PCR tested with oLC275 in combination with oLC3645 and oLC274 in combination with oLC3654. The *SAP2* promoter was induced to drive expression of FLP recombinase to excise the NAT flipper cassette. Once again, the NAT flipper cassette (pLC49)[60] was PCR amplified using primers oLC3641 in combination with oLC3642 and transformed into *SWI6/swi6::FRT*. NAT-resistant transformants were PCR tested as described above. The absence of a wild-type allele and the presence of the deleted allele was verified with oLC3645 in combination with oLC3431. The *SAP2* promoter was induced to drive expression of FLP recombinase to excise the NAT flipper cassette.

CaLC3391: To C terminally TAP tag Swi4, the TAP-*ARG4* cassette was PCR amplified from pLC573[62] using primers oLC3424 in combination with oLC3425 and transformed into CaLC239. Transformants prototrophic for arginine were PCR tested with oLC3426 in combination with oLC1593 and oLC1594 in combination with oLC3427.

CaLC3393: To C terminally TAP tag Swi6, the TAP-*ARG4* cassette was PCR amplified from pLC573[62] using primers oLC3428 in combination with oLC3429 and transformed into CaLC239. Transformants prototrophic for arginine were

PCR tested with oLC3430 in combination with oLC1593 and oLC1594 in combination with oLC3431.

**CaLC3395:** To C terminally TAP tag Swi4 in a strain with HA-tagged Cas5, the *TAP-ARG4* cassette was PCR amplified from pLC573[62] as described for CaLC3391 and transformed into CaLC3151. Transformants prototrophic for arginine and histidine were PCR tested as described for CaLC3391.

**CaLC3398:** To C terminally TAP tag Swi6 in a strain with HA-tagged Cas5, the *TAP-ARG4* cassette was PCR amplified from pLC573[62] as described for CaLC3393 and transformed into CaLC3151. Transformants prototrophic for arginine and histidine were PCR tested as described for CaLC3393.

**CaLC3932:** To generate a *GLC7* heterozygous deletion mutant, the NAT flipper cassette (pLC49)[60] was PCR amplified using primers oLC3490 in combination with oLC3491 and transformed into CaLC239 (SN95). NAT-resistant transformants were PCR tested with oLC275 in combination with oLC3558 and oLC274 in combination with oLC3561. The *SAP2* promoter was induced to drive expression of FLP recombinase to excise the NAT flipper cassette. To regulate the expression of *GLC7*, the tetracycline-repressible transactivator, the *tetO* promoter, and the NAT flipper cassette were PCR amplified from pLC605[61] using primers oLC3793 in combination with oLC3492 and transformed into *GLC7/glc7::FRT*. NAT-resistant transformants were PCR tested with oLC3794 in combination with oLC534 and oLC274 in combination with oLC3495. The absence of a wildtype *GLC7* promoter was verified with oLC3794 in combination with oLC3495, and the presence of a deleted allele was verified with oLC3794 in combination with oLC3494. The *SAP2* promoter was induced to drive expression of FLP recombinase to excise the NAT flipper cassette.

**CaLC3952:** To C terminally HA tag *CAS5* in the *GLC7* repressible strain, the *HA-HIS1* cassette was PCR amplified from pLC575[62] using primers oLC2161 in combination with oLC2162 and transformed into CaLC3932. Transformants prototrophic for histidine were PCR tested with oLC2163 in combination with oLC2164.

**CaLC3693/CaLC3694/CaLC3695:** To introduce a mutant allele of *CAS5* carrying S462E, S464E, S472E, and S476E mutations, the *CAS5S^{S462E/S464E/S472E/S476E}-HA-HIS1* cassette was released from pLC857 with *Sac*II and transformed into CaLC2034. Transformants prototrophic for histidine were PCR tested with oLC3052 in combination with oLC2029 and oLC2164 in combination with oLC1645. The absence of an untagged wild type *CAS5* allele was verified with oLC3052 in combination with oLC2164. The S462A/S464A/S472A/S476A mutations were sequence verified with oLC3371.

**CaLC3672/CaLC3673:** To introduce a mutant allele of *CAS5* carrying S462A, S464A, S472A, and S476A mutations, the *CAS5S^{462A/S464A/S472A/S476A}-HA-HIS1* cassette was released from pLC858 with *Sac*II and transformed into CaLC2034. Transformants prototrophic for histidine were PCR tested as described for CaLC3693. The absence of an untagged wildtype *CAS5* allele was verified with oLC3052 in combination with oLC2164. The S462A/S464A/S472A/S476A mutations were sequence verified with oLC3371.

**CaLC4471:** To C terminally TAP tag Swi4 in a mutant heterozygous for *SWI4*, the *TAP-ARG4* cassette was PCR amplified from pLC573[62] using primers oLC3424 in combination with oLC3425 and transformed into CaLC3151. Transformants prototrophic for arginine were PCR tested with oLC3426 in combination with oLC3427.

**CaLC4499:** To C terminally TAP tag Swi6 in a mutant heterozygous for *SWI6*, the *TAP-ARG4* cassette was PCR amplified from pLC573[62] using primers oLC3428 in combination with oLC3429 and transformed into *SWI6/swi6::FRT*. Transformants prototrophic for arginine were PCR tested with oLC3430 in combination with oLC1593 and oLC3430 in combination with oLC3431.

**CaLC4705:** To C terminally RFP tag Hhf1 in SN95, the RFP-NAT cassette was PCR amplified from pLC447 using primers oLC4752 in combination with oLC4753 and transformed into CaLC239. Correct integration at the C terminus of *HHF1* was verified by amplifying across both junctions using primer pairs oLC4417 in combination with oLC4754 and oLC274 in combination with oLC4755. To C terminally GFP tag Dad2, the GFP-HIS cassette was PCR amplified from pLC383[63] using primers oLC4748 in combination with oLC4749 and transformed into *HHF1-RFP/HHF1*. Correct integration at the C terminus of *DAD2* was verified by amplifying across both junctions using primer pairs oLC600 in combination with oLC4750 and oLC1645 in combination with oLC4751.

**CaLC4707:** To C terminally RFP tag Hhf1 in a *cas5Δ/cas5Δ* mutant, the RFP-NAT cassette was PCR amplified from pLC447 as described for CaLC4705 and transformed into CaLC2056. Correct integration at the C terminus of *HHF1* was verified as described for CaLC4705. To C terminally GFP tag Dad2, the GFP-HIS cassette was PCR amplified from pLC383[63] as described for CaLC4705 and transformed into *cas5Δ/cas5Δ HHF1-RFP/HHF1*. Correct integration at the C terminus of *DAD2* was verified by as described for CaLC4705.

**Plasmid construction.** Cloning procedures were performed following standard protocols. Plasmids used in this study are listed in Supplementary Table 2. The absence of nonsynonymous mutations in plasmids was verified by sequencing. Primers used in this study are listed in Supplementary Table 3. Plasmid construction is described in detail below.

**pLC447:** This is a construct to tag proteins with RFP at the C terminus. CaCherry/RFP was PCR amplified from pLC435[64] with oLC841 in combination with oLC842 and digested with BsrGI. This was cloned into pLC389 at BsrGI.

Directionality of the insert was verified by PCR with pLC849 in combination with oLC842. The construct was sequence verified with oLC849 and oLC842.

**pLC790:** This is a construct to C terminally HA-tag Cas5 in *C. albicans*. The *CAS5* DNA-binding domain, HA, and HIS marker were amplified from CaLC3113 genomic DNA with primers oLC3092 in combination with oLC3052 and cloned into pLC49 at the *Apa*I site. The ligation mixture was transformed into TOP10 cells. Cells were plated on LB + Amp + NAT. Correct integration of the cassette in pLC49 was sequence verified with oLC2029. This construct can be liberated by digestion with *Apa*I and transformed into *C. albicans*.

**pLC791:** This plasmid is based on pLC790 but harbors a phosphomimetic mutation in *CAS5* (S769E). This mutation was introduced by site-directed mutagenesis with primers oLC2986 and oLC2987. The clone was sequence verified with oLC2029. This construct can be liberated by digestion with *Apa*I and transformed into *C. albicans*.

**pLC800:** This plasmid is based on pLC790 but harbors a phosphomimetic mutation in *CAS5* (S772E). This mutation was introduced by site-directed mutagenesis with primers oLC2990 and oLC2991. The plasmid was sequence verified with oLC2029. This construct can be liberated by digestion with *Apa*I and transformed into *C. albicans*.

**pLC818:** This is a construct to C terminally HA-tag Cas5 in *C. albicans*. The *CAS5* ORF, HA, and HIS1 were amplified from CaLC3113 genomic DNA with primers oLC3365 in combination with oLC3366 and cloned into pLC49 at the *Sac*II site. The ligation mixture was transformed into TOP10 cells. Cells were plated on LB + Amp + NAT. Correct integration of the cassette in pLC49 was sequence verified with oLC2029, oLC244, and oLC3371. This construct can be liberated by digestion with *Sac*II and transformed into *C. albicans*.

**pLC828:** This plasmid is based on pLC818 but harbors a phosphomimetic mutation in *CAS5* (S472E/S476E). This mutation was introduced by site-directed mutagenesis with primers oLC3410 and oLC3411. The plasmid was sequence verified with oLC3371. This construct can be liberated by digestion with *Sac*II and transformed into *C. albicans*.

**pLC857:** This plasmid is based on pLC828 but harbors additional phosphomimetic mutations in *CAS5* (S462E/S464E). This mutation was introduced by site-directed mutagenesis with primers oLC3416 and oLC3417. The plasmid was sequence verified with oLC3371. This construct can be liberated by digestion with *Sac*II and transformed into *C. albicans*.

**pLC833:** This plasmid is based on pLC818 but harbors a phosphodeficient mutation in *CAS5* (S472A/S476A). This mutation was introduced by site-directed mutagenesis with primers oLC3412 and oLC3413. The clone was sequence verified with oLC3371. This construct can be liberated by digestion with *Sac*II and transformed into *C. albicans*.

**pLC858:** This plasmid is based on pLC833 but harbors additional phosphodeficient mutations in *CAS5* (S462A/S464A). This mutation was introduced by site-directed mutagenesis with primers oLC3418 and oLC3419. The plasmid was sequence verified with oLC3371. This construct can be liberated by digestion with *Sac*II and transformed into *C. albicans*.

**Culturing conditions.** All cultures were grown shaking at 200 rpm. To assess the effects of caspofungin and calcofluor white treatment on the expression of Cas5 and Cas5-dependent of transcripts, overnight cultures were diluted to $OD_{600}$ of 0.15 in YPD for 3 h, and either untreated or treated with 125 ng/ml of caspofungin or 100 μg/ml of calcofluor white for 1 or 2 h, as specified. To deplete *GLC7*, strains were grown overnight at 30 °C in YPD. Stationary phase cultures were split, adjusted to an $OD_{600}$ of 0.15, at which point one culture was treated with 0.02 μg/ml of doxycycline, whereas the other was left untreated. Cells were grown for 16 h and were split again, adjusted to an $OD_{600}$ of 0.15 and grown for 3 h. One culture was subsequently treated with 125 ng/ml of caspofungin for 1 or 2 h as specified, whereas the other was left untreated.

**ChIP-Seq data generation and analysis.** Chromatin preparation and immuno-precipitation against PolII using 2 μl of 8WG16 antibody were performed as described previously[65]. Multiplex sequencing libraries were constructed according to Wong et al.[66] ChIP-Seq libraries were assessed by Bioanalyzer DNA High Sensitivity Assay and quantified using real-time PCR before sequencing on the Illumina HiSeq2500 platform. Raw reads were mapped to the *C. albicans* reference genome. Mapped reads on gene bodies were counted, normalized to total number of mapped reads and expressed as "Normalized PolII ChIP-Seq signal". Differentially bound genes are defined by greater than 1.5-fold difference in PolII ChIP-Seq signals between any two conditions analyzed. Non-expressed genes or lowly expressed genes (e.g., genes with PolII ChIP-Seq signals lower than 10) under both conditions analyzed were removed from downstream analysis. Pathway analysis was carried out using the GO program on the *Candida* Genome Database (http://www.candidagenome.org/cgi-bin/GO/goTermFinder) and the KEGG mapper (http://www.genome.jp/kegg/tool/map_pathway2.html).

**Minimum inhibitory concentration assay.** Antifungal susceptibility was measured in flat bottom, 96-well microtitre plates (Sarstedt #83.3924) using a broth micro-dilution protocol described in ref. [67]. In brief, minimum inhibitory concentration (MIC) assays were set up in twofold serial dilutions of caspofungin or calcofluor white in a final volume of 200 μl per well. Caspofungin gradients were diluted

either from 125 ng/ml down to 0.12 ng/ml or 2000 to 3.9 ng/ml. Calcofluor white gradients were diluted from 250 μg/ml down to 0.24 μg/ml. Where applicable, doxycycline was added to a final concentration of 0.1 μg/ml. Cell densities of overnight cultures were determined and dilutions were prepared such that ~$10^3$ cells were inoculated into each well. Plates were incubated in the dark at 30 °C for 48 h, at which point the absorbance was determined at 600 nm using a spectrophotometer (Molecular Devices) and corrected for background from the corresponding medium. Each strain was tested in duplicate in three biological replicates. MIC data was quantitatively displayed with color using the program Java TreeView 1.1.3 (http://jtreeview.sourceforge.net).

**Flow cytometry analysis.** Yeast strains were cultured overnight in 4 ml YPD at 30 °C. Cells were subcultured for 4–5 h and isolated during exponential growth. Overall, 400 μl of cells were pelleted and fixed overnight in 70% ethanol at 4 °C. The fixed cells were washed twice with 1 ml of sodium citrate (50 mM) and resuspended in 500 μl sodium citrate and 12 μl of RNaseA (25 mg/ml). The cells were incubated at 37 °C for 4 h. Then 17 μl of propidium iodide was added to the RNase-treated cells and incubated overnight at 37 °C. The cells were vortexed and sonicated for 5 s at level 15 of the Sonic Desmembrator (Fisher Scientific), and filtered immediately before analysis using a BD Falcon tube with a 35 μm nylon mesh cap (Cat. No. 352235). Ploidy was determined using a Yetti flow cytometer, and 50,000–100,000 cells were analyzed per strain.

**Co-immunoprecipitation.** Cultures were grown to mid-exponential phase and harvested by centrifugation at 3000 rpm for 5 min. The cells were washed with sterile $H_2O$ and resuspended in 1 mL of lysis buffer (20 mM Tris pH 7.5, 100 mM KCl, 5 mM MgCl, and 20% glycerol), with one protease inhibitor cocktail per 50 ml (complete, EDTA-free tablet, Roche Diagnostics, Indianapolis, IN, USA) and 1 mM PMSF (EMD Chemicals, Gibbstown, NJ, USA). Cells were lysed by bead beating twice for 4 min with 7 min on ice between cycles. Lysates were recovered by piercing a hole in the bottom of each tube, placing each tube in a 14 ml conical tube, and centrifuging at 1300×g for three 5-min cycles, recovering the supernatant after each stage. The combined lysate was cleared by centrifugation at 21,000×g for 10 min at 4 °C. The protein concentrations were determined using the Bradford assay[68]. Anti-HA immunoprecipitations were done using Pierce HA-Tag IP/Co-IP kit (Thermo-Fisher Scientific, PI23610), as per manufacturer's instructions.

**Western blot analysis.** Protein was extracted by pelleting cells at $OD_{600}$ of 0.8, with the pellet being resuspended in 2× sample buffer (one-third volume of 6× sample buffer containing 0.35 M Tris-HCl, 10% (w/w) SDS, 36% glycerol, 5% β-mercaptoethanol, and 0.012% bromophenol blue). The samples were boiled for 5 min at 95 °C. The cell debris was pelleted and the supernatant was separated on a 6% SDS-PAGE gel to observe changes in Cas5 mobility. Separated proteins were electrotransferred to PVDF membrane (Bio-Rad Laboratories) and blocked with 5% skim milk in phosphate-buffered saline (PBS) with 0.2% Tween-20 at room temperature for 1 h. Blots were hybridized with antibody against the HA epitope (1:5000 dilution; Roche Diagnostics), p44/42 (1:2000, Cell Signaling), PSTAIRE (1:5000, Santa Cruz Biotechnology), TAP (1:5000; Open Biosystems), Hsp90 (1:10,000; generously provided by Brian Larsen, Des Moines University) or α-β-actin (1:5000; Santa Cruz, sc-47778) overnight at 4 °C. Blots were washed with PBS with 0.1% Tween-20 and subsequently hybridized with FITC-conjugated secondary antibody diluted 1:5000 in the block solution for 45 min at room temperature. Signals were detected using an ECL western blotting kit as per the manufacturer's instructions (Pierce).

**Two-dimensional western blot analysis.** *C. albicans* Cas5-HA cells were harvested from log phase cultures in YPD (+/−caspofungin) with lysis buffer (150 mM NaCl, 50 mM Tris-HCl pH 7, 15 mM EDTA, 1% Triton 100-X, 10% glycerol) plus phosphatase inhibitors (Sigma, P5726 and P0044) and protease inhibitors (Roche complete Mini Protease Inhibitor Cocktail Tablets 04693124001) using glass beads and bead beating 10 × 20 s with 1 min ice in between. Cell extracts were precipitated with methanol–chloroform and resuspended in rehydration buffer (Bio-Rad Cat#163-2105). Approximately 30 μg protein was loaded onto 7 cm IPG strips (pH 3-10, Bio-Rad Cat#163-2000) and resolved using Bio-Rad Protean i12 IEF System. The IPG strips were then resolved using 10% SDS-PAGE gels and processed for western blotting following the procedures described above for one-dimensional western blot analysis.

**Quantitative RT-PCR.** To prepare samples for RNA extraction, 10 ml of subculture was harvested by centrifugation at 1300×g for 5 min. The pellet was flash-frozen and stored at −80 °C overnight. RNA was isolated using the QIAGEN RNeasy kit and cDNA was generated using the AffinityScript cDNA synthesis kit (Stratagene). qRT-PCR was carried out using the Fast SYBR Green Master Mix (Thermo-Fisher Scientific) in 384-well plates with the following cycle conditions: 95 °C for 10 min, repeat 95 °C for 10 s, 60 °C for 30 s for 40 cycles. The melt curve was completed with the following cycle conditions: 95 °C for 10 s and 65 °C for 5 s with an increase of 0.5 °C per cycle up to 95 °C. All reactions were done in triplicate. Data were analyzed in the Bio-Rad CFX manager 3.1. Data was plotted using GraphPad Prism.

**Affinity purification and liquid chromatography-tandem mass spectrometry analysis.** For affinity purification (AP) and liquid chromatography-tandem mass spectrometry (LC-MS/MS) analysis, clarified whole-cell lysates (10 mg protein per sample) were extracted with immobilized anti-HA beads (Pierce HA-Tag IP/Co-IP kit, Thermo-Fisher Scientific, PI23610). After a 3-h incubation, the beads were washed three times with lysis buffer and twice with HPLC water. The proteins bound on beads were eluted with 150 μl of 0.15% trifluoroacetic acid (TFA), neutralized with 100 mM $NH_4HCO_3$, and digested with trypsin. The tryptic peptides were purified by 200 μl C18 stage tips (Thermo Scientific, Rockford, IL, USA) and analyzed by Q-Exactive LC-MS/MS[69]. The tryptic peptides from anti-HA IP complexes were separated on a 50-cm Easy-Spray column with a 75-μm inner diameter packed with 2 μm C18 resin (Thermo Scientific, Odense, Denmark). The peptides were eluted over 120 min (250 nl/min) using a 0 to 40% acetonitrile gradient in 0.1% formic acid with an EASY nLC 1000 chromatography system operating at 50 °C (Thermo-Fisher Scientific). The LC was coupled to a Q-Exactive mass spectrometer[70] by using a nano-ESI source (Thermo-Fisher Scientific). Mass spectra were acquired in a data-dependent mode with an automatic switch between a full scan and up to 10 data-dependent MS/MS scans. Target value for the full scan MS spectra was 1e6 with a maximum injection time of 120 ms and a resolution of 70,000 at $m/z$ 400. The ion target value for MS/MS was set to 1,000,000 with a maximum injection time of 120 ms and a resolution of 17,500 at $m/z$ 400. The first mass for the MS/MS was set to 140 $m/z$ and the normalized collision energy was set to 27. Unassigned, as well as charge states 1 and >5 were ignored for MS/MS selection. Repeat sequencing of peptides was kept to a minimum by dynamic exclusion of sequenced peptides for 12 s[71].

Acquired raw files were analyzed by using MaxQuant software[72] (version 1.3.0.5) for quantification, and X! Tandem (The GPM, thegpm.org; version CYCLONE; 2010.12.01.1) and Scaffold (version Scaffold_3.4.3, Proteome Software, Portland, OR, USA) for further validation. The default search parameters were used as described by Deeb et al.[71]. The search included cysteine carbamidomethylation as a fixed modification, as well as N-terminal acetylation, methionine oxidation, and phospho-serine, -threonine, and -tyrosine as variable modifications. Localization probability for phosphorylation sites was required to exceed 75%. The second peptide identification option in Andromeda was enabled. For statistical evaluation of the data obtained, the posterior error probability and false discovery rate were used. The false discovery rate was determined by searching a reverse database. A false discovery rate of 0.01 for proteins and peptides was permitted. Two miscleavages were allowed, and a minimum of seven amino acids per identified peptide were required. Peptide identification was based on a search with an initial mass deviation of the precursor ion of up to 6 ppm, and the allowed fragment mass deviation was set to 20 ppm. To match identifications across different replicates and adjacent fractions, the "match between runs" option in MaxQuant was enabled within a time window of 2 min.

**Microscopic analysis.** Cultures were grown to mid-exponential phase and 1 ml of the cells was centrifuged at 14,000 rpm for 1 min. The supernatant was removed and the cells were washed with PBS. To stain for chitin, calcofluor white was added to a final concentration of 1 μg/ml in a final volume of 100 μl. The cells were incubated in the dark for 15 min and were gently vortexted every 3–4 min. Cells were washed with PBS and 2 μl of the cells were deposited on a cover slide. To stain for nuclei, the cells were heat fixed on the slide for 1 min at 75 °C and 0.5 μl of 1 mg/ml DAPI was added. All imaging was performed on a Zeiss Imager M1 upright microscope and AxioCam Mrm with AxioVision 4.7 software. For fluorescence microscopy, an X-cite series 120 light source with ET green fluorescent protein (GFP), 4′,6-diamidino-2-phenylindole (DAPI) hybrid, and ET HQ tetra-methylrhodamine isothiocyanate (TRITC)/DsRED filter sets from Chroma Technology (Bellows Falls, VT, USA) was used. Calcofluor white and DAPI were viewed under the DAPI hybrid filter and HA-tagged Cas5 was viewed under the Texas Red filter. To facilitate nuclei counting, 12 Z stacks were taken for each image, where each slice is 0.3 μm.

**Indirect immunofluorescence microscopy.** To obtain cells for immuno-fluorescence, 5–10 ml of subculture was grown to mid-exponential phase and fixed with 5% formaldehyde for 3–4 h at 30 °C. Cells were harvested at 1300×g for 5 min and washed once with 5 ml S-Buffer (50 mM HEPES pH 7.5, 1.2 M Sorbitol) before being resuspended in 1 ml S-Buffer. To induce spheroplast formation, 10 μl of 1 M DTT, 30 μl of glusulase, and 40 μl of 2.5 mg/ml zymolase were added to the cells and the mix was incubated for 30 min at 37 °C. The extent of spheroplasting was monitored under the microscope. The poly lysine (Lys) coated slides were prepared by adding 15 μl of 0.1% poly Lys per slide well and set aside for 15 min. Wells were washed three times with PBS. The fixed cells were centrifuged for 1 min at 5000 rpm and gently resuspended in 1 ml S-buffer. Following the addition of 0.1% Triton X-100, the mix was incubated for 5 min on rocker. The cells were centrifuged for 1 min at 5000 rpm and resuspended in 1 ml S-Buffer. To adhere the cells, 20 μl of cell suspension was added on to poly Lys-coated well and incubated it for 15 min. The cells were washed three times with PBS-T and the wells were blocked with 20 μl PBS/BSA for 5–10 min. Anti-HA antibody diluted 1:300 in 20 μl was added and incubated for 3–4 h or overnight in a humid chamber. The cells were washed four times in PBS-T. Anti-mouse IgG-Cy3 diluted 1:500 in PBS-T was added to the wells and incubated for 1–2 h. The cells were washed four times with

PBS-T and once with only PBS. Overall, 20 μl of 1 mg/ml DAPI diluted 1:1000 in PBS was added to the wells and incubated for 5 min. The cells were washed four times with PBS. The cells were left to dry in the dark for 30 min at room temperature. Mounting medium was added and cells were viewed under the microscope.

**Data availability**. The ChIP-seq data have been deposited in the NCBI Sequence Read Archive with accession code SRP106998. The authors declare that all other data supporting the findings of the study are available in this article and its Supplementary Information files, or from the corresponding author upon request.

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

## Acknowledgements

We thank Cowen lab members for helpful discussions. We also thank David Rogers (University of Tennessee) for sharing microarray analysis of the *CAS5* homozygous mutant, and Li Ang (University of Macau) for assistance in optimizing the ChIP-Seq experiments. J.L.X. is supported by a Canadian Institutes of Health Research Doctoral award and M.D.L. is supported by a Sir Henry Wellcome Postdoctoral Fellowship (Wellcome Trust 096072). B.T.G. holds an Ontario Graduate Scholarship. C.B. and B.J.A. are supported by the Canadian Institutes of Health Research Foundation Grants (FDN-143264 and -143265). D.J.K. is supported by a National Institute of Allergy and Infectious Diseases grant (1R01AI098450) and J.D.L.C.D. is supported by the University of Rochester School of Dentistry and Medicine PREP program (R25 GM064133). A.S. is supported by the Creighton University and the Nebraska Department of Health and Human Services (LB506-2017-55). K.H.W. is supported by the Science and Technology Development Fund of Macau S.A.R. (FDCT; 085/2014/A2). L.E.C. is supported by the Canadian Institutes of Health Research Operating Grants (MOP-86452 and MOP-119520), the Natural Sciences and Engineering Council (NSERC) of Canada Discovery Grants (06261 and 462167), and an NSERC E.W.R. Steacie Memorial Fellowship (477598).

## Author contributions

J.L.X., T.K., M.F.M., D.J.K., C.B., B.J.A., A.S., K.H.W., N.R. and L.E.C. designed the experiments. J.L.X., B.T.G., J.D.L.C.D., K.T., M.D.L., T.K., A.S. and N.R. performed the experiments. L.Q., Z.M., K.T. and K.H.W. performed the RNA PolII ChIP experiments and data analysis. J.R.K., J.T. and M.F.M. performed AP and LC-MS analysis. J.L.X., N.R. and L.E.C. wrote the manuscript.

## Additional information

**Competing interests:** The authors declare no competing financial interests.

