## [Peer Review File · Nature Communications]

Reviewers' comments:

Reviewer #1 (Remarks to the Author):

This study is of the "landmark" class for *C. albicans*. It presents the mechanistic basis for the coupling of cell cycle and cell wall integrity regulation. The processes are both of exceptional interest in the fungal pathogenesis community, but their connection has been elusive for a variety of reasons. This study combines expression profiling, molecular/cell biological analysis, protein-protein interaction studies, and some very clever thinking to reveal that the novel *C. albicans* cell wall regulator, Cas5, interacts with the cell cycle regulator complex Swi4-Swi6. Beyond that, the authors discover (through clever deduction and some challenging genetic manipulations) a major upstream regulator of Cas5, the protein phosphatase Glc7. The story is exceptional from the standpoints of significance, scope, and superlative data quality.

The one aspect of the story that is a bit frustrating for this reader (and must have been for the authors) has to do with analysis of prospective phosphorylation site mutants. They clearly put a lot of work into this, but did not identify a specific phosphorylation site(s) that is critical for function. The S->E mutations that inactivate the protein may be acting as phosphomimics, but they may also be intolerable amino acid substitutions for other reasons. My view is that all of these mutations are presented efficiently, and there is the S residue in the DNA binding domain that has promising properties. Although this aspect of the story is not resolved, my feeling is that the amount of data in the manuscript is extensive, and further work on phospho-sites should be the subject of future studies.

One question - was mass spec analysis was carried out on caspofungin-treated cells as well as on untreated cells?

One suggestion - it would be interesting to see if GLC7 overexpression activates Cas5 in the absence of cell wall stress.

Reviewer #2 (Remarks to the Author):

This manuscript presents careful and comprehensive analyses of the function of the Cas5 transcription factor of *Candida albicans*. Cas5 is unique to *C. albicans* and does not have a *S. cerevisiae* counterpart, it was originally shown to be important for the cellular responses to echinocandin treatment and controlled a number of drug-responsive genes. Here the function has been extended to show that even in the absence of drug treatment Cas5 plays an important role in cells. Comparison of analyses of ChIP-seq of RNA polymerase II in control and mutant strains revealed that Cas5 has different regulatory circuitry dependent upon whether drug was present or absent. For the first time Cas5 was demonstrated to be dephosphorylated (and activated) by Glc7 phosphatase and Cas5-mediated control of cell wall homeostasis was partly dependent on the Swi4/Swi6 SBF complex. Another novel role for Cas5 in nuclear division control was uncovered altogether Cas5 is implicated in coordination of cell wall stress responses induced upon antifungal treatment and cell cycle. These findings are novel and will be of substantial interest to the molecular medical mycology research community. The experiments have been robustly performed and adequately validated.

Minor comments:

Line 121 amend to – The sets that had reduced RNA PolII occupancy upon deletion of Cas5 was significantly enriched in genes with functions in diverse processes.....and interactions with the host. In contrast, the gene set that had.....enriched in genes with functions associated with....

Line 175 - Figure 3b as the FKS1 point mutant/cas5 null strain grew marginally better than the cas5 null with increasing caspofungin treatment this would infer that there is some Cas5-

independent responses contributing to reduced susceptibility of the FKS1 point mutation. This should be mentioned in the text, Cas5 does not completely abrogate resistance acquired with the FKS1 point mutation.

Figure 4 and discussion of results on page 11. How much of the cell cycle regulated gene expression changes are related to inhibition of growth/killing of cells by the drug treatment (higher levels of G1-specific genes etc) and as the cas5 null mutant is hypersensitive to caspofungin could this result in the differences seen between the control strain and mutant?

Figure 5 the conclusions stated on page 13: Cas5 is activated by dephosphorylation in response to cell wall stress are not justified by the results presented here, you only show this later in the manuscript. From Figure 5 results all that can be deduced is that Cas5 is required for caspofungin-mediated upregulation of ECM331 and PGA13, you do not show that dephosphorylation is required in these experiments.

Lines 263/264: additional serines tested were S462 and S476. S464 is written twice.

Reviewer #3 (Remarks to the Author):

This manuscript provides new mechanistic information about the relationship among cell wall stress, drug resistance and cell cycle regulation in the major human fungal pathogen *Candida albicans*. More specifically, the authors demonstrate that the transcriptional regulator Cas5 is important for both responding to caspofungin, an echinocandin class antifungal that targets beta-glucan synthesis, as well as controlling the *C. albicans* cell cycle. Interestingly, ChIP-seq studies to examine PolII promoter occupancy indicate that Cas5 has distinct sets of target genes under basal vs. cell stress conditions. In addition, the authors demonstrate that dephosphorylation of Cas5 by the Glc7 phosphatase is important for the ability of Cas5 to control drug resistance and the cell cycle. Finally, the authors show that Cas5 physically associates with components of the SNP complex which, in turn, are important for target gene regulation.

In general, the manuscript is well-written and the data are clearly presented. The demonstration that control of cell wall stress responses, drug resistance and cell cycle regulation are coordinated by a common transcriptional regulator is a highly novel and significant finding that will be of general interest to a broad readership. In addition, because Cas5 lacks an ortholog in most other eukaryotes, this study highlights the potential for this key regulator to serve as an important target for the development of novel antifungal strategies. Since *C. albicans* is a major human fungal pathogen and only three major classes of antifungals are available, there is a significant demand to develop new therapies. Despite the strengths listed above, however, the manuscript also has a number of weaknesses, as indicated below:

1. While the authors do a good job in establishing the link between Glc7 phosphatase and Cas5 activity, it was unclear exactly how Glc7 itself was controlled in response to conditions of cell wall stress or drug treatment. Adding more mechanistic information about this important upstream part of the pathway would strengthen the manuscript.
2. Regarding Figure 5, do other cell wall stresses or does treatment with other antifungals besides caspofungin cause Cas5 to be localized to the nucleus? In addition, does treatment with azole drugs also effect dephosphorylation and activation of Cas5?
3. Lines 251-253 and lines 256-257: based on data in Figure 5i, the authors have not specifically demonstrated here that dephosphorylation of Cas5 is coupled to activation of gene expression (this demonstration occurs later in the paper). All that is shown here is that caspofungin-mediated induction of ECM331 and PGA13 is reduced in the cas5 mutant.
4. Line 263: based on Supplementary Figure 3, phosphorylation is detected at S462, not S464. There are also some additional concerns with this figure. In part b, the specific Cas5 residues that are mutated are not indicated. In addition, some of the data appears to be missing since

there should be four SA mutants and four SE mutants. Part c of this figure is also missing a control for WT CAS5 as well as a loading control.

5. It would be useful to show gene expression changes in Figure 4c for all the genes shown in Figure 4b, not just MCM2 and MCM3.

6. In Figure 4d why do genes in specific phases of the cell cycle show increased or decreased Pol II binding in response to caspofungin treatment? Some additional discussion of this interesting observation is warranted.

7. Westerns in Figures 5e, 5h, 6d, 7b and 7c are missing loading controls.

8. In order to further define the relationship between Cas5 and components of the SNP complex it would be useful to repeat experiments in Figures 8b, 8c as well as 9a and 9b using *cas5/cas5 swi4/swi4* and *cas5/cas5 swi6/swi6* homozygous double deletion mutants.

9. It would help to have a model figure at the end of the manuscript which clearly shows the multiple roles of Cas5 in controlling various biological processes as well as upstream regulation of Cas5 by the Glc7 phosphatase.

Reviewer #1 (Remarks to the Author):

This study is of the "landmark" class for C. albicans. It presents the mechanistic basis for the coupling of cell cycle and cell wall integrity regulation. The processes are both of exceptional interest in the fungal pathogenesis community, but their connection has been elusive for a variety of reasons. This study combines expression profiling, molecular/cell biological analysis, protein-protein interaction studies, and some very clever thinking to reveal that the novel C. albicans cell wall regulator, Cas5, interacts with the cell cycle regulator complex Swi4-Swi6. Beyond that, the authors discover (through clever deduction and some challenging genetic manipulations) a major upstream regulator of Cas5, the protein phosphatase Glc7. The story is exceptional from the standpoints of significance, scope, and superlative data quality.

Many thanks!!

The one aspect of the story that is a bit frustrating for this reader (and must have been for the authors) has to do with analysis of prospective phosphorylation site mutants. They clearly put a lot of work into this, but did not identify a specific phosphorylation site(s) that is critical for function. The S->E mutations that inactivate the protein may be acting as phosphomimics, but they may also be intolerable amino acid substitutions for other reasons. My view is that all of these mutations are presented efficiently, and there is the S residue in the DNA binding domain that has promising properties. Although this aspect of the story is not resolved, my feeling is that the amount of data in the manuscript is extensive, and further work on phospho-sites should be the subject of future studies.

We agree! Future studies that would be interesting to pursue include both investigating the phospho-sites important for caspofungin resistance and cell cycle regulation, as well as identifying the kinase important for Cas5 phosphorylation.

One question - was mass spec analysis was carried out on caspofungin-treated cells as well as on untreated cells?

We did perform mass spec analysis on caspofungin treated cells and on untreated cells. However, the dephosphorylated form of Cas5 precipitates under non-denaturing conditions, precluding inclusion of this sample in our analysis.

One suggestion - it would be interesting to see if GLC7 overexpression activates Cas5 in the absence of cell wall stress.

We show in the absence of doxycycline *GLC7* is overexpressed in the *tetO-GLC7/glc7Δ* strain (Figure S4). As shown in Figure 7, Cas5 is not activated under these conditions, suggesting that transcriptional overexpression of *GLC7* is not sufficient to activate Cas5 in the absence of cell wall stress. In future studies, it would be interesting to determine if even stronger overexpression of *GLC7* would be sufficient to cause constitutive activation of Cas5.

Reviewer #2 (Remarks to the Author):

This manuscript presents careful and comprehensive analyses of the function of the Cas5 transcription factor of Candida albicans. Cas5 is unique to C. albicans and does not have a S. cerevisiae counterpart, it was originally shown to be important for the cellular responses to echinocandin treatment and controlled a number of drug-responsive genes. Here the function has been extended to show that even in the absence of drug treatment Cas5 plays an important role in cells. Comparison of analyses of ChIP-seq of RNA polymerase II in control and mutant strains revealed that Cas5 has different regulatory circuitry dependent upon whether drug was present or absent. For the first time Cas5 was demonstrated to be dephosphorylated (and activated) by Glc7 phosphatase and Cas5-mediated control of cell wall homeostasis was partly dependent on the Swi4/Swi6 SBF complex. Another novel role for Cas5 in nuclear division control was uncovered altogether Cas5 is implicated in coordination of cell wall stress responses induced upon antifungal treatment and cell cycle. These findings are novel and will be of substantial interest to the molecular medical mycology research community. The experiments have been robustly performed and adequately validated.

Many thanks!

Minor comments:

Line 121 amend to – The sets that had reduced RNA PolII occupancy upon deletion of Cas5 was significantly enriched in genes with functions in diverse processes.....and interactions with the host. In contrast, the gene set that had.....enriched in genes with functions associated with....

We have made the indicated changes to the text. (Now lines 122-126).

Line 175 - Figure 3b as the FKS1 point mutant/cas5 null strain grew marginally better than the cas5 null with increasing caspofungin treatment this would infer that there is some Cas5-independent responses contributing to reduced susceptibility of the FKS1 point mutation. This should be mentioned in the text, Cas5 does not completely abrogate resistance acquired with the FKS1 point mutation.

We have altered the text to reflect the reduction in echinocandin resistance that was observed as opposed to complete abrogation of resistance. (Now line 178).

Figure 4 and discussion of results on page 11. How much of the cell cycle regulated gene expression changes are related to inhibition of growth/killing of cells by the drug treatment (higher levels of G1-specific genes etc) and as the cas5 null mutant is hypersensitive to caspofungin could this result in the differences seen between the control strain and mutant?

Results from our studies as well from studies with *S. cerevisiae* suggest that alteration in cell cycle dynamics in response to cell wall stress is a tightly coordinated and regulated process that enables cells to survive and respond to environmental insult. These responses require specific cellular regulators, and although they are not specific to cell wall stress, they do not reflect a

signature of cell death. For example, in *S. cerevisiae*, plasma membrane stress inhibits S-phase entry through a mechanism that depends on Mck1-dependent degradation of Cdc6, a component of the pre-replicative complex (our ref 53; Kono et al. 2016. PNAS. 113: 6910-6915); this mechanism enables survival, as continued DNA synthesis and cell cycle progression in the presence of plasma membrane damage induces membrane rupture and cell lysis.

Our results suggest that Cas5 has important roles in repressing Cdc6 as well as other members of the MCM complex (Figure 4b and 4c) in response to caspofungin. This is consistent with a model in which in the absence of Cas5, Cdc6 is not repressed, thereby enabling cell cycle progression and ultimately resulting in cell death, as defects in cell wall homeostasis and integrity are lethal during continued cell growth and division. We have elaborated on this point in our Discussion (Lines 394-396 and lines 401-402) and have added a model (Fig. 10) to our revised submission.

Figure 5 the conclusions stated on page 13: Cas5 is activated by dephosphorylation in response to cell wall stress are not justified by the results presented here, you only show this later in the manuscript. From Figure 5 results all that can be deduced is that Cas5 is required for caspofungin-mediated upregulation of ECM331 and PGA13, you do not show that dephosphorylation is required in these experiments.

We have revised our conclusion to state: “Thus, Cas5 is regulated by dephosphorylation and governs the expression of caspofungin-responsive cell wall genes in response to cell wall stress.” Lines 259-260.

Lines 263/264: additional serines tested were S462 and S476. S464 is written twice.

Many thanks for catching this error. We have corrected the text. (Line 266).

Reviewer #3 (Remarks to the Author):

*This manuscript provides new mechanistic information about the relationship among cell wall stress, drug resistance and cell cycle regulation in the major human fungal pathogen *Candida albicans*. More specifically, the authors demonstrate that the transcriptional regulator Cas5 is important for both responding to caspofungin, an echinocandin class antifungal that targets beta-glucan synthesis, as well as controlling the *C. albicans* cell cycle. Interestingly, ChIP-seq studies to examine PolII promoter occupancy indicate that Cas5 has distinct sets of target genes under basal vs. cell stress conditions. In addition, the authors demonstrate that dephosphorylation of Cas5 by the Glc7 phosphatase is important for the ability of Cas5 to control drug resistance and the cell cycle. Finally, the authors show that Cas5 physically associates with components of the SNP complex which, in turn, are important for target gene regulation.*

In general, the manuscript is well-written and the data are clearly presented. The demonstration that control of cell wall stress responses, drug resistance and cell cycle regulation are coordinated by a common transcriptional regulator is a highly novel and significant finding that will be of general interest to a broad readership. In addition, because Cas5 lacks an ortholog in most other eukaryotes, this study highlights the potential for this key regulator to serve as an

*important target for the development of novel antifungal strategies. Since *C. albicans* is a major human fungal pathogen and only three major classes of antifungals are available, there is a significant demand to develop new therapies.*

Many thanks!

Despite the strengths listed above, however, the manuscript also has a number of weaknesses, as indicated below:

1. While the authors do a good job in establishing the link between Glc7 phosphatase and Cas5 activity, it was unclear exactly how Glc7 itself was controlled in response to conditions of cell wall stress or drug treatment. Adding more mechanistic information about this important upstream part of the pathway would strengthen the manuscript.

We agree that investigating the manner by which Glc7 is regulated in *C. albicans* would be an interesting avenue to pursue in the future. However, given the extensive amount of data included in this manuscript that outlines the mechanistic basis by which Cas5 couples the regulation of cell cycle and cell wall integrity, we think further experiments investigating Glc7 regulation are beyond the scope of this study.

2. Regarding Figure 5, do other cell wall stresses or does treatment with other antifungals besides caspofungin cause Cas5 to be localized to the nucleus? In addition, does treatment with azole drugs also effect dephosphorylation and activation of Cas5?

In Figure 5a we demonstrate that treatment with the cell wall stressor caspofungin results in Cas5 nuclear translocation. This treatment is identical to the treatment in Figure 5d and 5e that shows Cas5 is induced and dephosphorylated in response to caspofungin, leading to the model that dephosphorylation of Cas5 results in its nuclear translocation to mediate transcriptional responses important for cell wall stress. Figure 5e and 5f confirm that other forms of cell wall stress result in dephosphorylation of Cas5, highlighting that this response is not specific to caspofungin.

We have attempted some preliminary experiments with fluconazole and have observed that although Cas5 appears to be dephosphorylated in response to this antifungal its levels are not induced, highlighting a potential divergence in the role of Cas5 in responding to distinct environmental stressors. Given our focus of the role of Cas5 in responding to cell wall stress, experiments exploring its role in responding to cell membrane stress are beyond the scope of this study.

3. Lines 251-253 and lines 256-257: based on data in Figure 5i, the authors have not specifically demonstrated here that dephosphorylation of Cas5 is coupled to activation of gene expression (this demonstration occurs later in the paper). All that is shown here is that caspofungin-mediated induction of ECM331 and PGA13 is reduced in the cas5 mutant.

We have revised our conclusion to state: “Thus, Cas5 is regulated by dephosphorylation and governs the expression of caspofungin-responsive cell wall genes in response to cell wall stress.” (Lines 259-260).

4. Line 263: based on Supplementary Figure 3, phosphorylation is detected at S462, not S464. There are also some additional concerns with this figure. In part b, the specific Cas5 residues that are mutated are not indicated. In addition, some of the data appears to be missing since there should be four SA mutants and four SE mutants. Part c of this figure is also missing a control for WT CAS5 as well as a loading control.

We have revised the text to indicate that phosphorylation was detected at residue S462 (Line 266).

In part b, the mutant that was generated consists of all four serine residues (S462, S464, S472, and S476) mutagenized to either glutamic acid, to mimic a constitutively phosphorylated state, or alanine, to mimic a constitutively unphosphorylated state in the same strain. This is described on Lines 266-269. We have also revised the Supplemental Figure 3 legend to be more clear about the strain being tested. Given that no phenotype was observed when all four residues were mutated, we did not test each residue individually.

In part c, the western blot is comparing a Cas5 mutant mimicking a constitutively phosphorylated state to that mutant to mimicking a constitutively unphosphorylated state, with no differences observed between the phosphoshift in these strains in response to caspofungin. We do not make reference to the wild-type strain, which is why it was not included in the Figure. It is Cas5 mobility not protein levels that is relevant for this experiment, and therefore a loading control is not required.

5. It would be useful to show gene expression changes in Figure 4c for all the genes shown in Figure 4b, not just MCM2 and MCM3.

Our current RT-PCR data validates the RNA-PolIII chip seq and there is little value added by inclusion of a few additional genes. We have made sure our text reflects the data that is shown and only comment on expression levels of *MCM2* and *MCM3*. Lines 209-211.

6. In Figure 4d why do genes in specific phases of the cell cycle show increased or decreased Pol II binding in response to caspofungin treatment? Some additional discussion of this interesting observation is warranted.

We agree that the link between cell wall stress and cell cycle arrest is an interesting observation. We have highlighted in the Discussion that “this is the first mechanistic insight into the signaling events that couple cell cycle arrest with cell wall stress in *C. albicans*.” (Lines 397-398). In terms of why specific phases of the cell cycle show increased or decreased binding, this appears to be reminiscent of what has been observed in *S. cerevisiae* in which plasma membrane stress inhibits S-phase entry (our ref 53; Kono et al. 2016. PNAS. 113: 6910-6915); this mechanism enables survival, as continued DNA synthesis and cell cycle progression in the presence of plasma

membrane damage induces membrane rupture and cell lysis. We have elaborated on this point in our Discussion (Lines 399-401 and lines 406-407).

7. Westerns in Figures 5e, 5h, 6d, 7b and 7c are missing loading controls.

Given that Figure 7b concludes Cas5 levels are induced in response to caspofungin regardless of transcriptional repression of *GLC7*, we have now included a loading control for this Figure and appreciate the reviewer noting this omission.

All other Western blots are monitoring the phosphoshift of Cas5 in response to cell wall stress. Cas5 levels are not pertinent for these experiments, and are not discussed in the manuscript, and thus a loading control is not required.

8. In order to further define the relationship between Cas5 and components of the SNP complex it would be useful to repeat experiments in Figures 8b, 8c as well as 9a and 9b using cas5/cas5 swi4/swi4 and cas5/cas5 swi6/swi6 homozygous double deletion mutants.

We agree that further investigation into the relationship between Cas5 and Swi4/Swi6 would be an interesting avenue to pursue in the future. However, given the extensive amount of data included in this manuscript that outlines the mechanistic basis by which Cas5 couples cell cycle and cell wall integrity regulation, the suggested experiments are beyond the scope of this manuscript.

9. It would help to have a model figure at the end of the manuscript which clearly shows the multiple roles of Cas5 in controlling various biological processes as well as upstream regulation of Cas5 by the Glc7 phosphatase.

We think this is an excellent suggestion that will both summarize and clarify the roles of Cas5 in orchestrating cell cycle dynamics in response the cell wall stress. We have included a model (Fig. 10) in the revised manuscript.

REVIEWERS' COMMENTS:

Reviewer #1 (Remarks to the Author):

My comments have been addressed perfectly well. This story remains terrific!

Reviewer #2 (Remarks to the Author):

The authors have done an excellent job of responding to all comments made by the reviewers.

Reviewer #3 (Remarks to the Author):

In general, the authors have satisfactorily addressed my previous concerns as well as concerns of the other reviewers and the manuscript is significantly improved. The revised manuscript is also now strengthened by a new model figure which clearly depicts the role of Cas5 in controlling cell cycle progression and cell wall stress response, in addition to regulation by the Glc7 phosphatase. There are only two minor points remaining:

1. The response to the first question in my previous second point, about whether other cell wall stresses or treatment with other antifungals besides caspofungin causes nuclear localization of Cas5 (referring to Figure 5) was not complete. While it is appreciated that other forms of cell wall stress lead to dephosphorylation of Cas5 (Figures 5e and 5f) and nuclear translocation is implied, no data was shown to specifically demonstrate Cas5 nuclear localization under these conditions.
2. The "b" part label for Figure 4b is missing.

Reviewer #1 (Remarks to the Author):

My comments have been addressed perfectly well. This story remains terrific!

Many thanks for both your kind words and original comments that helped us to improve the clarity and impact of our manuscript.

Reviewer #2 (Remarks to the Author):

The authors have done an excellent job of responding to all comments made by the reviewers.

Many thanks for both your kind words and original comments that helped us to improve the clarity and impact of our manuscript.

Reviewer #3 (Remarks to the Author):

In general, the authors have satisfactorily addressed my previous concerns as well as concerns of the other reviewers and the manuscript is significantly improved. The revised manuscript is also now strengthened by a new model figure which clearly depicts the role of Cas5 in controlling cell cycle progression and cell wall stress response, in addition to regulation by the Glc7 phosphatase. There are only two minor points remaining:

Many thanks! Your original comments assisted in us improving the overall impact and clarity of our manuscript.

1. The response to the first question in my previous second point, about whether other cell wall stresses or treatment with other antifungals besides caspofungin causes nuclear localization of Cas5 (referring to Figure 5) was not complete. While it is appreciated that other forms of cell wall stress lead to dephosphorylation of Cas5 (Figures 5e and 5f) and nuclear translocation is implied, no data was shown to specifically demonstrate Cas5 nuclear localization under these conditions.

We have revised the text to be more accurate in our conclusions. Line 268-269 now reads: “Thus, in response to cell wall perturbation by caspofungin, Cas5 is induced and translocates to the nucleus.”

2. The “b” part label for Figure 4b is missing.

We have added the label to the Figure.